# Review of Applications of Density Functional Theory (DFT) Quantum Mechanical Calculations to Study the High-Pressure Polymorphs of Organic Crystalline Materials

**DOI:** 10.3390/ijms241814155

**Published:** 2023-09-15

**Authors:** Ewa Napiórkowska, Katarzyna Milcarz, Łukasz Szeleszczuk

**Affiliations:** Department of Organic and Physical Chemistry, Faculty of Pharmacy, Medical University of Warsaw, Banacha 1, 02-093 Warsaw, Poland; enapiorkowska@wum.edu.pl (E.N.); kmilcarz@wum.edu.pl (K.M.)

**Keywords:** pressure-induced phase transition, DFT calculations, polymorphism, solid state

## Abstract

Since its inception, chemistry has been predominated by the use of temperature to generate or change materials, but applications of pressure of more than a few tens of atmospheres for such purposes have been rarely observed. However, pressure is a very effective thermodynamic variable that is increasingly used to generate new materials or alter the properties of existing ones. As computational approaches designed to simulate the solid state are normally tuned using structural data at ambient pressure, applying them to high-pressure issues is a highly challenging test of their validity from a computational standpoint. However, the use of quantum chemical calculations, typically at the level of density functional theory (DFT), has repeatedly been shown to be a great tool that can be used to both predict properties that can be later confirmed by experimenters and to explain, at the molecular level, the observations of high-pressure experiments. This article’s main goal is to compile, analyze, and synthesize the findings of works addressing the use of DFT in the context of molecular crystals subjected to high-pressure conditions in order to give a general overview of the possibilities offered by these state-of-the-art calculations.

## 1. Introduction

With decades of usage by physicists and mineralogists, high-pressure science is an established field. With the aid of improvements in experimental methods, high-pressure research has expanded over the years from these roots into a variety of diverse domains. The field of crystal engineering is showing increasing interest in and research on the behavior of organic and metal–organic systems under high pressure.

The use of temperature to create or alter materials has dominated chemistry from its very beginnings, while pressure of more than a few tens of atmospheres is only seldom encountered. However, pressure is a highly potent thermodynamic variable that is increasingly applied to molecular systems to obtain new materials or modify existing ones.

Depending on the topic of study, the term “high pressure” might refer to a number of different variables. On the scale employed by many planetary scientists, what is extraordinary for biologists interested in life at large marine depths hardly qualifies. In the field of crystal engineering, the range between approximately 0.1 and 20 GPa is usually described as “high pressure”. It should be noted that the difference between atmospheric pressure and 1 GPa represents a factor of 10^4^, which is an enormous value for molecular materials, where we can typically alter the temperature only within a few hundred degrees. Large structural changes in molecular solids may be brought about by such pressure as contributions made by pressure-volume terms (pV) to free energy in this range can be equal to covalent bond energies [1].

Numerous publications discuss the polymorphic changes caused by pressure in molecular solids. Sometimes, the high-pressure polymorph and the low-temperature form are identical, but this is not always the case. Moreover, high-pressure polymorphs may have the same space group symmetry as the original ambient-pressure form and even be isostructural with it, which is defined as an isosymmetric phase transition. The only way to distinguish between a phase transition and an anisotropic structural distortion is by direct optical inspection of the interface and a series of structure refinements at various pressure values that demonstrate whether or not the cell parameters and volume changes with pressure are continuous [2].

Since intermolecular potentials, atomic pseudopotentials, and other computational protocols used to model the solid state are typically optimized with reference to ambient-pressure structural data, their application to high-pressure problems is a very difficult test of their validity from a computational point of view. Nevertheless, the application of quantum chemical calculations, usually at the level of density functional theory (DFT), has been proven multiple times to be an excellent tool that can be used not only to explain, at the molecular level, the observations of high-pressure experiments [3], but also to predict properties that can be further validated by experimentalists [4].

The main aim of this article is to gather, discuss, and summarize the results of the studies addressing the application of DFT in the field of molecular crystals subjected to high-pressure conditions to provide an overview of the possibilities offered by state-of-the-art calculations. Due to the popularity and availability of DFT codes enabling such calculations, the complete list of organic, inorganic, and metal–organic compounds that have been studied using computational methods under high pressure is enormous. To describe all of these cases, a full-length book would be required, which is beyond the scope of this review. Therefore, we limit the reviewed works to those describing solid organics, following the title of this Special Issue.

This review is divided into sections. After the Introduction (Section 1), the main part begins with a brief summary of the concepts of DFT calculations (Section 2), followed by the presentation of the published results and a discussion of the most interesting, state-of-the-art examples, grouped by the properties and applications of the studied compounds (Section 3). Then, the following section presents the results dealing with fundamental properties such as structure, enthalpy, vibrations, and phonons (Section 4), followed by a part devoted to particular problems, e.g., anisotropic compression (Section 5), and finally a section describing the general conclusions (Section 6).

## 2. Brief Theoretical Background

Atomistic model simulations based on DFT are undoubtedly the most popular quantum chemical methods for the analysis of the structure–property relationships of crystalline compounds [5,6,7]. Almost 60 years ago, by demonstrating that the total energy of electrons moving in an external potential is a distinct functional of the electron density, Hohenberg and Kohn (1964) [8] created the underlying formalism of DFT. Consequently, by decreasing the total energy with regard to the electron density, the ground state of the quantum mechanical system of electrons in a solid may be determined. Kohn and Sham (1965) [9] then proposed a workable strategy for DFT computations, requiring the self-consistent solution of the so-called Kohn–Sham (KS) equations. The KS equations translate an external potential system of n interacting electrons into an effective potential system of n noninteracting electrons to describe an external potential system of n interacting electrons. The Born–Oppenheimer approximation, which isolates the dynamics of the electrons from those of the considerably heavier nuclei, is used in the vast majority of implementations. DFT has been very popular for calculations in solid-state physics since the 1970s. However, the rapid development and application of this method in modeling molecular crystals started in the 1990s [10].

The type of functional and a basis set are the two most important variables that must be chosen before starting the DFT calculations, and this choice greatly affects the obtained results, as shown in Section 3.3.1 describing the studies of solid chlorothiazide. Functionals characterize the electronic energy quantitatively, which, when combined with the kinetic and electrostatic energy of the system, yields the system’s overall energy. In studies devoted to organic solids, the most popular DFT functional is undoubtedly the PBE [11] (Table 1). The choice of the basis set depends on the code being used. A range of software basis sets (Table 2).

Van der Waals interactions and hydrogen bonds, among other noncovalent forces, are essential for the creation, stability, and operation of the majority of solid organics. At present, high-level quantum chemical wavefunctions or the quantum Monte Carlo technique are the only ways to adequately account for omnipresent van der Waals interactions. All prominent local-density or gradient-adjusted exchange-correlation functionals of DFT, as well as the Hartree–Fock (HF) approximation, on the other hand, lack the appropriate long-range interaction tail. The Kohn–Sham DFT equation does not take into account a long-range electron correlation effect, sometimes referred to as the London component of the dispersion energy factor. This problem has a long history of negatively impacting the DFT calculations’ accuracy. There are several dispersion correction techniques accessible today. They should be examined independently for each system of concern because their inclusion does not always increase the calculation accuracy. Tkatchenko–Scheffler (TS) [90], Grimme Dispersion (GD, often written as D) [91], and Many-Body Dispersion (MBD) [92] are the three most used dispersion corrections in the study of solid organics.

The most frequent activity in all of the possible molecular simulations is the optimization or minimization (with respect to potential energy) of the system being examined. The ground-state structure of a compound is often calculated by minimizing the total energy with regard to the locations of the atoms and, in the case of crystalline systems, also the lattice parameters (a, b, c, α, β, γ) in athermal conditions, also omitting the zero point motion (ZPVE) and at 0 GPa. This approach is usually called “full geometry optimization”. The computation of structures at high pressure (but still within the athermal limit) is then based on minimizing the enthalpy with respect to a non-zero stress tensor [93]. 

There are several methods to account for the impact of temperature in model calculations, but each one needs a large amount of computing power. The free energy of crystalline solids may be expressed using the quasi-harmonic approximation, in which phonons do not interact with one another but the phonon frequencies depend on the unit cell volume [94].

Molecular dynamics simulations are an alternate method to QHA that yield results that go beyond the harmonic approximation. Conceptually, DFT-based molecular dynamics simulations are identical to molecular-mechanics-based MD simulations in that the forces acting on the atoms in a particular configuration are estimated, the atom locations are then updated in accordance with the selected time step, and this process is continuously repeated [95].

The above paragraph only briefly summarizes the basic concepts of DFT calculations and their practical implementations in describing molecular crystals, especially at high pressure. For readers interested in more detailed information on the conceptual backgrounds of such calculations, we recommend the more general, excellent review articles [3,93,96,97,98,99].

## 3. The Classes of Modeled Systems

The variety of modeled systems is enormous and there are multiple ways to group them. In this review, we gather the materials into three main classes: (1) organic materials with metal additives; (2) high-energy materials; and (3) pharmaceuticals. The first class is on the border between inorganic materials, which are not the focus of this review, and typical organic solids. Due to the presence of metals, the variety of bonds and interactions existing in these materials is very large, which makes them interesting from a structural point of view but also quite difficult to model. The second group is important in the context of high pressure, which is usually connected with high-energy materials. The third group has emerged as polymorphism that plays a key role in pharmaceutical sciences due to the differences in stability, dissolution, and processability between the particular polymorphs of active pharmaceutical ingredients.

### 3.1. Organic Materials with Metal Additives

Understanding the responses of organometallic materials to pressure changes is essential in optimizing their performance and designing new materials with tailored properties. In silico methods provide a valuable tool for the investigation of the effects of pressure on organic materials with metal additives, allowing for the efficient and cost-effective exploration of a wide range of pressure conditions. The application of DFT methods has appeared in many works as a useful tool in such cases (Table 1).

#### 3.1.1. Methylammonium Lead Bromide (MAPbBr_3_)

Among the group of organic materials with the addition of metals, MAPbBr_3_ has attracted major attention. The DFT periodic calculations are known to be applicable in describing the electronic properties of the structure. Therefore, the purpose of the first research [14] was to investigate the structural flexibility of MAPbBr_3_, which provides a platform to engineer the optoelectronic properties by adjusting the bandgap energies by applying external pressure or temperature. The band-edge calculations predicted the bandgap to be direct up to 2 GPa, which is eligible in photovoltaics applications. According to the results, the calculated bandgap energies were lower than the experimental values, which is a well-known drawback of the PBE functional. Nevertheless, the absolute values of the bandgap energies were not crucial for this analysis. More importantly, the reduction of the bandgap with pressure observed in the DFT calculations was in agreement with the experimental pressure-dependent photoluminescence energies in the cubic *I* ($Pm\bar{3}m$) phase.

The authors of the other work [17] provided an experimental investigation of MAPbBr_3_ under various stress conditions supported by DFT calculations. First-principles molecular dynamics (FPMD) simulations were performed to demonstrate the effect of the CH_3_NH_3_ (MA) organic molecules on the instantaneous and time-averaged bandgaps of the various structural phases. NVT FPMD simulations were carried out for pressure of 0.1/0.2/0.4 GPa, 0.6/1.0/1.4 GPa, and 1.8/2.4/3.0 GPa, on the phases $Pm\bar{3}m$, *Im*$\bar{3}$, *Pnma*, respectively. DFT studies showed that the dynamics of the MA organic molecule and the inorganic lattice, which were connected by N−H···Br hydrogen-bonding interactions, affected the distance between the Pb and Br and the bandgap evolution under pressure. This study also highlighted the importance of MD simulations in obtaining good-quality results. According to the static (at 0 K) calculation results, the bandgap of the *Pnma* phase at 0 K was predicted to decrease as a result of pressure-induced band broadening. However, the bandgap obtained by averaging the outcomes from snapshots of the FPMD simulations at 300 K increased with increasing pressure, which was in agreement with the experimental observations. The combination of experimental and theoretical investigations provided an insight into the behavior of MAPbBr_3_ under pressure, which can be useful in the construction of optoelectronic devices. 

The aim of the next work [16] was to investigate band structure evolution on the compression of halide perovskites like MAPbBr_3_; see Figure 1. According to calculations at ambient pressure and confirmed experimentally, the investigated perovskites were direct bandgap semiconductors. With the increase in pressure, bandgap contraction was observed up to the first critical pressure at 4 GPa. Beyond this pressure value, the widening of the bandgap happened and the bandgaps of organic perovskites became indirect. With further compression, the bandgap remained direct; however, a decrease in the bandgap was observed. The calculated results for MAPbBr_3_ were in good agreement with the experimental findings, as was also the case in the earlier described works. With the help of pressure, the magnitude and the direct–indirect nature of bandgaps can be adjusted. It was pointed out that direct semiconductors are likely to have longer carrier lifetimes than indirect ones, which is a desired property for materials in photovoltaic applications.

#### 3.1.2. Methylammonium Lead Iodide (MAPbI_3_)

In the next work [18], a similar material to the previously mentioned one was studied. DFT calculations using hybrid functional B3PW91 with spin–orbit coupling (SOC) were applied to compare the bandgap changes of MAPbI_3_ polymorphs under compression with those obtained from laser-excited photoluminescence (PL) spectra. The calculated pressure-dependent bandgaps decreased from 1.67 to 1.62 eV up to 0.3 GPa and then increased from 1.58 to 1.74 eV between 0.4 and 2.0 GPa, which was in excellent agreement with the experimental values. It was noticed that the transformation from *I*4*/mcm* to *Im*$\bar{3}$ led to a smaller bandgap energy, whereas conversion from the *Im*$\bar{3}$ polymorph into the suggested *Immm* polymorph reversed this tendency. Since photoluminescence is directly related to polymorphism, conversion from the low-pressure *I*4*/mcm* phase to the cubic *Im*$\bar{3}$ supercell (at 0.4 GPa) provided a distinct procedure for the reduction of MAPbI_3_ bandgaps and may be useful in the development of materials in photovoltaic applications.

Another work [15] also focused on the structures of MAPbI_3_ under various external pressure values. In the study, the structures of various phases of this material were optimized and their electronic as well as optical properties were calculated. Calculations included the density of states and partial density of states, band structures and projection band structures, and the changes in charge transfer amount and electronic charge density. At the phase transition point, the obtained bandgaps of these structures were higher than their experimental values (error range: 8%). Based on the results of this work, it was possible to evaluate changes in optical properties due to pressure-induced phase transition.

#### 3.1.3. Chloroindium(III) Hybrid Perovskite (IPy)_4_[In_2_Cl]_10_

In the next work [12], the authors investigated the structural and optical characterization of a chloroindium(III) hybrid perovskite (IPy)_4_[In_2_Cl]_10_ at high pressure. According to the experimental part of the study, no phase transitions were observed up to 1.51 GPa. However, several changes in electronic and structural properties were observed. Based on these findings, computational methods could help to understand the pressure effect on the noncovalent interactions and band structure, and thus the properties, of hybrid perovskites.

#### 3.1.4. Zn(μ-Cl)_2_(3,5-Dichloropyridine)_2_]_n_

The authors of the next work [13] performed an incremental simulation of the hydrostatic compression and decompression of the plastically flexible coordination polymer [Zn(μ-Cl)_2_(3,5-dichloropyridine)_2_]_n_. The tetragonal space group symmetry was conserved for anisotropic compression simulations, with the possibility of full cell relaxation, wherein an external potential was placed along the crystallographic c-axis. The anisotropic compression was performed to study how the compression direction affected the observed phase transition. A series of hydrostatic compression simulations were performed with a pressure step of 0.5 GPa and no symmetry restrictions. At around 5 Gpa, a structural phase transition was observed, although the pressure-volume (p-V) curves did not show discontinuities, typical for phase transformation. The DFT calculations revealed that this transition was caused by the pressure-induced softening of low-frequency vibrations and it was associated with breaking symmetry from $P\bar{4}b2$ to $P\bar{4}$. It was stated that the calculated unit cell volume of the experimental ambient-pressure structure was overestimated by less than 2%; thus, it was well represented by PBE TS. The results for compression and decompression simulations were in good agreement with experimental data and they showed that the phase transformation was predictable and reversible.

#### 3.1.5. Pt(bpy)Cl_2_

The authors of the next article [19] applied DFT calculations to investigate the red-to-yellow phase transition experimentally observed in Pt(bpy)Cl_2_ microcrystals at high pressure. To simulate the yellow phase, the monomer was used, but to simulate the stacked linear-chain red form, dimer and trimer units were employed. The theoretical approach helped to analyze the experimentally observed differences in the Pt–Pt distance and electronic structures manifested in differences in the optical spectroscopic properties of the two phases. The presence of hysteresis in the optical properties allowed them to infer that the yellow phase obtained on compression was not the same as observed at ambient pressure. According to the calculations, the high-pressure luminescence corresponding to the high-pressure yellow phase was more likely a halide–ligand transition rather than a ligand–field transition. The authors concluded that supporting experimental methods with theoretical work can be beneficial in the explanation of pressure-induced phase transitions.

### 3.2. High-Energetic Organic Materials 

Studying high-energetic materials at various pressure values is crucial for several reasons. Primarily, pressure has a significant impact on the stability, reactivity, and performance of these materials. Detonation and deflagration expose energetic materials to extreme conditions, resulting in vital transformations in their properties. These changes impact various characteristics, including their susceptibility to shock initiation and their chemical reactivity [100]. Additionally, studying high-energetic materials under different pressure values can provide valuable insights into their structural changes, phase transitions, and energetic properties, enabling the development of safer and more efficient energetic materials.

#### 3.2.1. Silver Fulminate (AgCNO)

The authors of the next work [34] investigated the relative thermodynamic phase stability of two polymorphs of silver fulminate (AgCNO) at high pressure and in a wide range of temperatures (Figure 2). The research of energetic materials like AgCNO is challenging due to their sensitivity to extreme conditions and risk of decomposition. Hence, in silico simulations help to predict the physicochemical properties of these materials under high pressure and temperatures. In this study, the authors used two plane wave DFT codes, namely PWSCF and CASTEP, which is quite unusual as, in most of the other works, the authors usually do not compare different programs. In addition, the authors checked how the choice of the DFT functional and empirical dispersion correction affected the results of calculations. This was beneficial as, according to the study’s results, the pressure-induced phase transition from β to α at 2.5 GPa was observed only when the geometry optimization was performed without any dispersion correction (GGA PBE). These results were in contrast to those obtained using the DFT-D2 approach, indicating that the α-form was more stable in the whole range of studied pressure values. Moreover, the calculated volumes of the α and β forms were overestimated by 22.5% and 13.5% with GGA PBE, for each phase, respectively. The application of DFT-D2 methods resulted in minor discrepancies of approximately 2.2% and 4.5% for the α and β form, respectively. These obtained results emphasize the need to include dispersion correction, which, in some cases, is crucial to accurately model structural properties and their pressure dependence. Further, the bond length and bond angle were evaluated up to 5 GPa using DFT-D2 methods to characterize the compressibility behaviors of the polymorphs. It was noticed that the polymorphs behaved anisotropically under hydrostatic pressure. The calculated electronic band structure using the TB-mBJ functional at ambient pressure showed that the α and β phases were indirect bandgap insulators, with bandgap values of 3.51 and 4.43 eV, respectively. More examples of the investigation of high-energy materials and the need to study them are described in the section below.

#### 3.2.2. 3,5-Trinitrohexahydro-S-Triazine (RDX)

A few studies on RDX with the application of DFT methods have been published. One of the works [69] investigated the compression behaviors of α, γ, and ε forms of RDX and the impact of pressure on the heat capacity of RDX using phonon calculations. However, the results suggested very weak dependence in the case of all crystal forms of RDX, showing very small discontinuity at the α–γ phase transition. Furthermore, the authors suggested that modeling pressure–heat capacity dependence is challenging and would require a very sensitive technique. On the other hand, the obtained results of the lattice parameters of α, γ were in good agreement with previous computational DFT-D studies and experimental data. In this paper, the behavior of a high-temperature/high-pressure polymorph, the ε-polymorph, was also analyzed by a computational compression study for the first time and the obtained results were in good accordance with the experimental lattice parameters, with the largest deviation from the experimental volume being 0.9%. The geometry optimizations were performed by maintaining the crystal space groups. 

In another work [22], the authors used DFT calculation to describe the differences in the intermolecular interactions and conformational changes of RDX under high pressure. The combination of experimental measurements and calculations indicated another phase transition of β-RDX at 6.4 GPa induced by the conformation variation of β-RDX. 

In the next work [30], the authors calculated the Gibbs free energy to obtain the relative stability of forms α and γ of RDX at a pressure range of 0–10 GPa and temperature up to 450 K. Based on the results, the transition pressure from form α to form γ was suggested to be temperature-independent, as, in the interval of range 50–450 K, it was equal to 3.8 GPa, which was in good accordance with the experimental outcomes.

#### 3.2.3. 2,6-Diamino-3,5-Dinitropyrazine-1-Oxide (LLM-105)

Another commonly studied material is LLM-105. In the first work about this compound, the authors used DFT calculations to support the experimental investigation of the behavior of LLM-105, a nitro-energetic material, under high pressure [24]. Up to 25.7 GPa, the first-order phase transformation was not observed. The DOS and band structure of LLM-105 were calculated to determine the second-order phase transition at 10.5 GPa. The combination of theoretical and experimental approaches yielded that an electronic structure phase transition occurred at ca. 10 GPa and was associated with a sudden change in electronic transfer from the exposed oxygen atom to the amino groups. At around 25 GPa, the abrupt bandgap reduction suggested a pressure-induced phase transition and this was consistent with the experimental part of the work. 

In the second work [31], it was also stated that up to 20 GPa, there was no indication of a pressure-induced phase transition of LLM-105, even though the set of NPT molecular dynamic simulations was performed at various pressure conditions to obtain the pressure dependence on the lattice parameters. Nevertheless, at around 13 GPa, a significant shift in the compressibility of the b-axis relative to the other axes was noticed, which was attributable to a decrease in the spacing between molecules in adjacent in-plane sheets. The authors concluded their work with a statement that EOS calculations can be used for advancements in modeling high-pressure–temperature detonation reactions of energetic materials such as LLM-105.

In the last study in this section [32], it was pointed out that the inclusion of dispersion correction is crucial to accurately model systems like LLM-105. Geometry optimization of the crystal structure at ambient pressure using the PW91 functional without dispersion correction generated a ca. 27% error for the unit cell volume, mostly due to the larger errors like 23.6% for the b-edge. However, using dispersion-corrected DFT methods allowed the authors to obtain results in very good agreement with the experiments. The experimental unit cell dimensions were well modeled up to 6 GPa, with a maximum deviation of approximately 0.05 Å. Above 6 GPa, there were no available experimental data for this moment to compare. There was no evidence of a structural phase transition up to 45 GPa as well in this work. However, small symmetry changes and fluctuations in the structural parameters were observed. 

#### 3.2.4. Cyclic Aliphatic Nitramine Octahydro-1,3,5,7-Tetranitro-1,3,5,7-Tetrazocin (HMX)

Another high-energetic material, HMX, has also obtained significant interest within the literature [27]. The authors of one paper studied changes in the α-polymorph of HMX under the influence of compression. They evaluated the pressure dependence on the hydrogen-bonding network based on geometric evolution, electronic structure, Hirshfeld surfaces, the method of atoms in molecules, the method of intramolecular gradients, the interpenetration distance, mechanical properties, and vibrational spectra. The molecular structure, crystal packing, lattice constants, volume, and bandgaps changed with increasing pressure, and intramolecular interactions were dominated by hydrogen bonds. The results suggested that the performance of the metastable α-HMX crystal is induced by pressure-triggered hydrogen bond strengthening, and the intermolecular interactions clearly influence the mechanical properties of the crystal. From phonon dispersion curves, a large negative frequency at 4 GPa showed the dynamical instability of α-HMX at high pressure, which was in line with experimental studies.

In another work [33], the pressure-induced metallization of condensed phase β-HMX using quantum molecular dynamics in conjunction with a multi-scale shock technique (MSST) was investigated and a self-consistent charge density functional tight binding (SCC-DFTB) method was adopted. Based on simulations, a remarkable reduction in the bandgap was observed with the increasing pressure exerted by the shock wave front, and the phase became metallic at ca. 130 GPa, as a result of the breakage of the chemical bond, and chemical chain reactions were initiated. The major gaseous products of the decomposition of β-HMX were in agreement with experimental and previous theoretical results.

#### 3.2.5. 1,1-Diamino-2,2-Dinitroethene (FOX-7)

Another high-energy material appearing in the literature is FOX-7 [25]. The authors of one of the studies examined FOX-7 using dispersion-corrected density functional theory under uniaxial compression along three major crystalline directions. Calculations included principal and shear stresses as well as Raman spectra prediction to explain the role of hydrogen bonding in stability and phase transitions under pressure. The simulated Raman spectra showed anomalous changes attributed to the rearrangement of inter- and intramolecular structures, the alternation of bond lengths, and the rotation of groups. The findings of this work showed that non-hydrostatic pressure in comparison to hydrostatic compression could trigger a phase transition, possibly along different crystallographic directions. This work shows promising insights into studies performed on energetic materials such as FOX-7 in the case of hydrogen bonding, anisotropy, and sensitivity to shock initiation.

In the second study [29], the authors investigated the structural response of an insensitive energetic crystal of FOX-7 up to 12.8 GPa using synchrotron single-crystal X-ray diffraction measurements and DFT calculations (Figure 3). The comparison of the experimental unit volumes and angles with theoretical findings showed that they were reproduced well with the DFT-D approach, within a 1.3% error for the investigated pressure range. The lattice parameters were also well modeled, both below and above 4.5 GPa, with abrupt changes at the transition pressure. It was noticed that the predicted length along the axis normal to the molecular layers (precisely the b-axis below 4.5 GPa and a-axis above 4.5 GPa) varied from the measured values more than the lengths along the other two axes. Based on these findings, it was confirmed that even using the DFT-D approach did not reproduce well the weak interlayer interactions as well as strong in-layer interactions. Due to the increase in strength interactions at higher pressure, the agreement between the experimental values improved significantly. Overall, the results demonstrated that structural changes in the FOX-7 structure were reproduced well by DFT-D calculations of various phases and at the applied pressure range. The authors demonstrated that the chemical and structural stability of FOX-7 was mainly controlled by the accommodation of external compression through the anisotropic compressibility of the unit cell, phase transformation of the structure to planar layers, and an increase in the hydrogen and C−NO_2_ bonding strength. The authors stated that the results can be beneficial for a further understanding of the shock responses of insensitive, highly explosive crystals.

#### 3.2.6. 2,4,6-Trinitrotoluene (TNT)

Konar et al. studied two polymorphs of 2,4,6-trinitrotoluene (TNT), namely m-TNT and o-TNT, with the use of the PBE functional and two dispersion corrections, PBE TS and PBE-D2 [26], to evaluate the agreement between the experimental and corresponding calculated results. The results showed that the PBE-D2 method outperformed PBE TS in the case of a negligible temperature effect. At ambient conditions, though, PBE TS showed an advantage over PBE-D2. These results suggested that the choice of dispersion correction may depend on the specific pressure and temperature conditions of the system being studied.

#### 3.2.7. Pentazolates

In the next work [21], DFT calculations were used to investigate the geometry and crystal structures, electronic features, hydrogen-bonding network, and vibrational properties of two energetic pentazolate anion salts up to 50 GPa. Studying crystal structures’ behavior and their properties under extreme conditions provides guidelines for the application of pentazolates as energetic materials. Based on the results, in the case of (N_5_^−^)_2_DABTT_2_^+^, the critical point was found at 9 GPa, indicating the possibility of a phase transition at this pressure value. Furthermore, for N_5_GU^+^, it was not found. The observed distortions of the cations of both compounds were investigated, as they could affect the hydrogen-bonding network in the crystals and can have an impact on their stability. From the gradually decreasing bandgap under increasing pressure, it was noticed that the pressure enhanced the charge overlap and delocalization, thus improving the electronic transition from occupied orbitals to empty orbitals.

#### 3.2.8. 2,4,6-Trinitro-3-Bromoanisole (TNBA)

In another work [20], the authors applied the first-principles evolutionary crystal structure prediction method USPEX to evaluate the possible polymorphs of 2,4,6-trinitro-3-bromoanisole (TNBA), an energetic material, at 10 GPa. In order to find the most stable structures, the search was performed without any lattice or symmetry constraints, allowing the structure to be flexible. Although the validation search was performed for a structure stable at 0 GPa, this ambient phase crystal structure was not found. However, the structures that were obtained at 0 GPa were energetically competitive, with the difference from the ambient pressure phase being lower than 1 kcal/mol. An unreacted equation of state (EOS) for the ambient phase of TNBA was found; with the optimization of the lattice parameters, atomic configurations, and hydrostatic pressure, an isothermal method (at T = 0 K) was performed and the result was presented in the form of the function of the volumetric compression ratio V/V_0_. Calculations were performed for equilibrium volume V_0_ at ambient pressure and the reduction in the volume was sequential by 0.02 V_0_. The authors stated that the crystal structure prediction (CSP) simulations concerned decomposition products; however, they did not calculate the energy barrier, which is important in understanding the chemistry of energetic materials.

#### 3.2.9. Hexanitrohexaazaisowurtzitane (HNIW or CL-20)

Another article focused on the polymorphs of CL-20 [37]. The β and ε-CL-20 unit cells were optimized in CASTEP. However, due to the large size of α· H_2_O phase crystals, it was beyond the computational capabilities of the authors to perform its geometry optimization at that time (in 2007) and the γ-phase could not be optimized for unknown reasons. Hence, the single-point calculations of four forms were performed using the DMol3 software. Nevertheless, based on bandgap calculations, the experimental order of phases was accurately predicted. The least sensitive form (ε-CL-20) was used for further analysis under hydrostatic pressure. At 0 GPa, the calculated lattice constants were 1.5% or so larger than the measured ones, which is typical for the GGA-PBE method. At low and high pressure, the compressibility of ε-CL-20 is anisotropic; up to 10 GPa, minor changes in the lattice parameters, band structure, and DOS were observed. However, from 50 to 400 GPa, they changed greatly. The increment in pressure reduced the bandgap and consequently increased the sensitivity of ε-CL-20. Based on these changes, 400 GPa was considered to be the critical pressure for the insulator–metal phase transition.

#### 3.2.10. Triaminotrinitrobenzene (TATB)

In the next study [28], the authors applied DFT calculations to simulate Raman-active modes and compared theoretical models with experimental spectra of TATB. The simulated crystal structure at ambient pressure, as well as its volume–pressure dependence and Raman spectra, using DFT methods with implemented dispersion correction and thermal and zero-point energy contributions, were in good agreement with experimental findings. Furthermore, the calculated peak positions were in excellent agreement with experimental measurements from both this study and those performed by other researchers [101], with the highest inaccuracy of 10% in the low frequencies and less than 3% elsewhere. Up to 27 GPa, no first-order transition was observed as no discontinuities within the low-frequency mode evolution were detected, which was in accordance with the experimental study [28].

#### 3.2.11. RDX, HMX, CL-20, NM, TATB, and PETN

Most of the earlier mentioned energetic materials were also studied in the next work [35]. The authors performed a theoretical investigation using DFT calculations at ambient pressure and on compression and compared the calculated results with the experimental findings. In addition, sets of calculations were performed to test the impact of applying dispersion correction and a selected energy cut-off from 25 to 80 Ry on the accuracy of the predicted lattice parameters at ambient pressure. It was also emphasized that the level of agreement was highly improved by using the DFT-D method instead of the conventional DFT. Overall, good agreement between the calculated and experimental structures’ parameters was obtained using dispersion-corrected calculations as the results were within a 2% error range. Afterward, the hydrostatic compression effects on α and γ-RDX, β-HMX, ε-CL20, NM, TATB, and PETN crystals were explored at a wide pressure range selected to match the available experimental data, e.g., a range of 0–3.36 GPa for α-RDX and 3.9–7.99 GPa for γ-RDX. Similarly, in this case, including dispersion correction in the calculations improved the obtained results. On this basis, the authors claimed that the DFT-D method was capable of reproducing the changes in lattice parameters and unit cell volume caused by hydrostatic compression as the obtained maximum errors for the lattice parameters were 1.8% (α -RDX), 1.07% (γ-RDX), 3.67% (β-HMX), 0.91% (ε-HNIW), 2.62% (NM), 1.74% (TATB), and 2.79% (PETN), respectively, relative to compression data obtained at ambient temperature.

### 3.3. Pharmaceuticals

#### 3.3.1. Chlorothiazide

It has been noticed that phase transition can be kinetically hindered and applying pressure in static calculations may not be sufficient to detect it, even if the relative stability of polymorphs could suggest that the phase transformation should occur at the investigated conditions. To overcome this issue, several methods have been applied in theoretical approaches. It is worth mentioning that these approaches are usually very computationally demanding and their application can be limited by the size of the crystal’s unit cells to small or moderate-sized systems.

One of the solutions to the aforementioned problem can be the application of ab-initio molecular dynamics (aiMD) to include the kinetic energy factor in the phase transition [39]. For example, based on experimental findings, chlorothiazide, a diuretic agent, has been proven to undergo pressure-induced isosymmetric structural phase transition (IPT) of Form I to Form II at 4.2 GPa (Figure 4) [102]. The authors of the work [39] performed geometry optimization calculations of Form I and Form II at all experimentally studied pressure conditions to simulate compression (when geometry optimization starts from Form I) or decompression (when geometry optimization starts from Form II) using two functionals: PBE TS and PBESOL. The next stage was to measure the root mean square deviation (RMSD) values between the estimated and experimental crystal structures in order to assess the correctness of the modeled structures. According to the energy results obtained using the PBE TS functional, Form II ought to be more stable at higher pressure. However, the energy barrier prevented the IPT from being observed during geometry optimization. In the case of the PBESOL, Form II was preserved only for pressure higher than 3.5 GPa, and a jump discontinuity in the lattice parameters would suggest that, at this point, one can expect IPT. However, according to the PBESOL calculations, Form I should be more energetically favorable than Form II in the whole studied pressure range, which is in contrast to experimental observations and indicates that the dispersion correction is crucial for the accurate modeling of solid-phase transition. Hence, only the PBE TS was used for the thermodynamic and aiMD calculations. According to the variations in the predicted thermodynamic parameters (ΔH and TS), the pressure-induced IPT should be an entropy-driven transition. According to the computed changes in the Gibbs free energy (ΔG), the investigated forms should coexist at pressures between 3.5 and 4.1 GPa and, above this pressure, Form II should be more stable and dominant, which is in agreement with the experimental data. Therefore, by adding kinetic energy in the aiMD simulations, it was possible for the studied system (Form I) to overcome this energy barrier and reach the deeper minimum (Form II). This approach is very computationally demanding but allows us to overcome kinetic barriers. 

To conclude, the geometry optimization allowed the researchers to observe the pressure-induced IPT of Form II to Form I during decompression. The thermodynamic parameters indicated the possibility of reversed transformation and determined the relative stability of the forms at higher pressure. Based on the promising results of the static calculations, the authors successfully applied aiMD to overcome energy barriers and observe the IPT of Form I to Form I during the simulation.

#### 3.3.2. Urea

In the next work [45], the application of aiMD was described in the case of urea (Figure 5). The authors chose two approaches to predict the phase transition of urea between Form I and IV, occurring at 3.1 GPa. Firstly, they performed calculations of geometry optimization and thermodynamic calculations at 0 and 3.1 GPa. They stated that the accuracy of the data depended on the chosen functional. At lower pressure, calculations performed on Form IV as the initial structure fit the results obtained when the initial form was Form I, with good accuracy. A change in the crystal space group was observed as in the experiment, as well as the final unit cell dimensions. The values of energy and free energy, which were obtained in initializing calculations from different structures, were also almost identical. However, geometry optimization performed at higher pressure did not result in a change in the space group of Form I and the final structure differed significantly from the one that was obtained when starting from Form IV, but the values of differences in the energy and Helmholtz free energy between both structures correctly predicted their stability. The second approach involved the application of NPT quantum molecular dynamic calculations at 3.1 GPa. The received unit cell lengths of Form I equaled those for Form IV, as well as the final space group. The application of QMD calculations was stated to be a successful method in the study of the phenomenon of polymorphic phase transitions and a useful tool for accurate crystal structure prediction at different pressure values.

#### 3.3.3. Tolazamide

The polymorphic transition of tolazamide, an anti-diabetic drug, is another example of when a solid-state transformation is kinetically hindered. The authors of [51] investigated the pressure dependence on two forms of tolazamide (Form I and II) and the transformation to a denser form on compression using DFT calculations and experimental methods. The transition of polymorph I into Form II was not observed with increasing pressure, neither experimentally nor computationally. Furthermore, Form II did not transform into Form I, even though Form I was determined to be denser at all pressure values discussed in this work. The authors proposed two hypotheses to explain these observations. First, there was no thermodynamic-driven force due to the similarity in the free energies of polymorphs at the studied pressure. Second, Form I was more thermodynamically stable; however, the solid-phase transition was kinetically controlled. To test the former, the geometry optimization of two polymorphs was performed up to 20 Gpa. From the results of calculations, Form I had lower internal crystal energy and enthalpy than Form II at all simulated pressure values. The estimated enthalpy difference between the two phases was 6.1 kJ/mol and 96.8 kJ/mol at ambient pressure and 20 GPa, respectively. Interestingly, the zero-point energy (ZPE) contribution to the difference between the enthalpies did not exceed 4 kJ/mol and thus did not affect the obtained results. The discrepancy between the calculated and experimental outcomes could have occurred due to the lack of an entropy contribution in the calculated values. Nevertheless, the calculated results showed that Form I was more stable across the entire studied pressure range; thus, no transformation of Form I to Form II should be expected. On the other hand, the transformation of Form II to Form I could be possible if there was no kinetic barrier. Further investigation of the pressure impact on the structures of tolazamide revealed that the intermolecular interactions changed significantly under pressure. Comparison of the conformations of the polymorphs showed that the main difference was associated with the change in the C4–S1–C1–C8 dihedral angle; thus, tolazamide was assumed to be a conformational polymorph. The calculated value of the energy barrier of the transformation of one conformer into another was 28 kJ/mol, and this was in agreement with previously reported data. In a solid state, the estimated energy barrier could be even higher due to the intermolecular changes required to obtain conformational changes. Under ambient conditions, the barrier can easily be overcome in solutions, which was confirmed by the observation of the recrystallization of Form II into Form I in dioxane and methanol. Additionally, it was noted that increasing the temperature at ambient pressure facilitated the kinetically hindered phase transformation of the tolazamide polymorphs.

#### 3.3.4. Aspirin

The calculation of the energy barrier can help to justify the lack of experimentally observed polymorphic transition under pressure, as in the case of aspirin crystals. The behavior of these crystals under pressure was the area of interest in the next work [52]. The authors used dispersion-corrected DFT calculations to establish the relative stabilities of two forms of aspirin (Form I and Form II) and the shear–slip mechanism of their interconversion. Based on the result, Form I was predicted to be 0.3 kJ/mol per molecule less stable than Form II with B86bPBE-XDM. Including ZPE slightly decreased the difference by 0.1 kJ/mol, which was in reasonable agreement with previous studies using the PBE TS and PBE MBD approaches [103]. The predicted free-energy (ΔG) difference of 0.3 kJ/mol per molecule at 298 K, as thermal effects were included, favored Form I as being more stable than Form II, corresponding to the experimental preference. From the results of the PES scan for the phase transition from Form II to Form I, the obtained energy barrier of the transition increased from 10 kJ/mol per molecule at 0 GPa to the maximum value of ca. 18 kJ/mol per molecule at 7 GPa [52]. The results explained the lack of transition from Form II to I, supported by experimental work up to 10 GPa within limited timescales [104].

#### 3.3.5. Triclabendazole

As a pressure increase can be insufficient to observe a phase transition, in some cases, thermodynamics calculations should be performed to define the conditions required for the occurrence of phase transformations [30].

In [38], the authors calculated the pressure-dependent crystal structures of triclabendazole, an oral anthelmintic. In this work, instead of periodic DFT calculations, the authors used a “supercell” approach by constructing the 3 × 3 × 3 supercells of two polymorphs. By comparing the stabilities of two types of TCBZ using the DFT and embedded fragment techniques and Gibbs free energies comparison, they discovered a phase transition between TCBZ Forms I and II at 5.5 GPa and ≈350 K. Unfortunately, no such examination of the pressure-induced phase transition of TCBZ has been previously performed so as to be compared with the calculated results.

#### 3.3.6. Resorcinol

In the next work [41], the authors have used an unusual approach to describe the behavior of resorcinol under compression and high temperatures. It was presented that DFT and DFTB can be coupled in a quasi-harmonic model to correctly present the crystal structures, phase boundaries, and thermochemistry of the α and β phases of resorcinol. The authors suggested that the combination of DFT and DFTB+ for phonon calculations could provide a good balance between the computational time and accuracy of the results and reduce the calculation time by 1–2 orders of magnitude compared to the more conventional approach based solely on DFT. Although the use of DFTB3-D3(BJ) alone provided inaccurate results since, at all investigated temperature and pressure values, α-resorcinol was more thermodynamically stable and no α→β phase transition was observed, the mixed approach indicates the correct phase behavior in resorcinol. In this case, the results of Γ point DFT phonon calculations are comparable to the mixed approach. However, more significant differences could be expected in more complex examples with greater differences in conformation between polymorphic forms. The revolutionary approach is based on the phonon density of states calculations in a large supercell with DFTB and then shifting individual DFTB phonon bands to the frequency obtained from DFT calculations performed in a crystallographic unit cell. As a result, this operation ensures DFT-quality phonon modes at the Γ point, while the phonon dispersion is modeled with DFTB. Furthermore, the application of this mixed DFT/DFTB+ method creates the possibility to compute the thermodynamic properties for larger and more complex systems in a reasonable time. The QHA B86bPBE-XDM method has been applied to test the pressure dependence of the lattice constants. In the case of the α phase, the largest mean absolute error of 1.1% occurs for the b-edge and 1.3% for the a-constant for β-resorcinol, with a good representation of the experimental data. This model most accurately predicts the lattice parameters, while, at low pressures, the B86bPBE-XDM and PBE TS methods without quasi-harmonic thermal expansion underestimate the volumes [54]. Nevertheless, at high pressure, all approaches overestimate the volumes. From the phase diagram, the authors estimated the phase transition temperature in the 0–1 GPa regime. At ambient pressure, the phase transformation was predicted to occur at 368 K, corresponding almost exactly to the experimental conditions and at 260 K at 0.4 GPa, which is in reasonable agreement with observations at room temperature. It is noted that the phase transition of resorcinol strongly depends on temperature and pressure as a kinetic factor plays an important role in this phase transformation [41].

#### 3.3.7. Glycine

In the next work [42], the authors investigated the structural, electronic, and thermodynamic properties of the α-, β-, and γ-glycine polymorphs. The lattice parameters and the unit cell volume of the α phase was predicted with a maximum error of 1.42% and 2.53%, respectively. Based solely on the results of the lattice energy values, the stability order of the glycine polymorphs was in disagreement with the experimental data. However, the application of quasi-harmonic calculations allowed them to obtain the correct stability order. Excellent agreement was found for the isothermal volume behavior of the α-glycine in the 0 to 50 GPa range. The phase transition of the γ- to α-glycine was recognized at pressure values of 0.98 and 0.55 GPa at 300 and 400 K, respectively, and the α phase was indicated to be the most stable at 500 K within the investigated pressure range. The best predicted value of the γ-to-α-phase transition condition was obtained from a fit to the third-order Birch−Murnaghan equation of stat and they were 444.55 K and 1 bar, whereas, experimentally, it was around 440 K. This example has shown that entropy and thermal contribution could be crucial to accurately foresee the relative stability of the polymorphic forms and justify the need for Gibbs free energy calculation. In the case of glycine, the authors also revealed that the calculated order of phases could depend on the dispersion correction method as the experimental ranking of the lattice energy was obtained by using MBD dispersion [105] correction, although it was not achieved by using the D3 scheme or TS correction [106]. 

## 4. Fundamental Aspects of DFT Calculations at High Pressure

After the description of some of the most illustrative examples in the previous section, in this one, a different approach to results presentation is used. Instead of dividing the reviewed works into particular classes of compounds, below, the fundamental aspects of DFT calculations are described, starting from structure prediction and optimization through the calculation of vibrations, which are the essential part of the crystal entropy and the derivation of the thermodynamic properties. It should be noted here that the solid-state normal mode calculations are the another level of work compared to the geometry/crystal cell parameters’ optimization, both in terms of theory and the complexity of calculations and computational time needed to obtain the results. Finally, the calculations of phonon properties are described. 

### 4.1. Geomtry Optimization at Various Pressure Conditions and Crystal Structure Prediction

As in other types of studies utilizing molecular modeling methods, in the high-pressure DFT calculations geometry optimization is a requirement for the further calculation of various properties. Usually, the SCXRD-determined unit cell is used as the initial basis for further calculations. However, in the absence of an experimentally determined structure, the crystal structure prediction methods (CSP) can be used, which are described below.

The goal of the CSP is to determine the crystal structure of a solid based on the molecular structure. Computational methods that can be used to achieve this goal include simulated annealing, evolutionary algorithms, distributed multipole analysis, random sampling, basin hopping, data mining DFT, and molecular mechanics calculations. For many years, the CSP have been investigated as an addition to experimental solid form screening, assisting in the structural characterization of observed solid forms, understanding molecular crystallization conditions, and determining when it may be reasonable to stop screening [107,108]. Moreover, the CSP blind tests—a type of of CSP competition—are very popular not only among academics, with six editions already successfully completed [109] and the results of the seventh anticipated to be published [110].

In particular, the CSP methods based on dispersion-corrected density functional theory (DFT-D), available in the GRACE software, have been successfully used to model multiple compounds, whose behavior has been then studied also at high pressure [111,112], and this approach has proven to be very helpful in explaining experimental findings. More such examples are presented below.

Sometimes, in the case of high-pressure studies, the crystal structure determined experimentally at ambient conditions is then optimized at higher pressure, which can result in major structural changes. The presence of structural phase transition can be observed by abrupt changes in lattice parameters or volume on compression. The authors of the next work [43] investigated the isosymmetric first-order phase transition of L-histidine. The DFT calculations were performed with symmetry constraints. As a result, the discontinuities in the lattice parameters and unit cell volume showed the formation of new phases I′ and II′. The theoretical phase transformation for the orthorhombic form (I) was observed between 4.45 and 4.62 GPa, which is in good agreement with the experimental transition pressure equal to 4.5 GPa. In the case of the monoclinic phase, a reduction in volume was observed ca. between 3.00 and 3.21 GPa and experimentally at 3.1 GPa. 

Discontinuities in lattice parameters were also observed in the next work [65] where the authors investigated the bis-1,2,3-thiaselenazolyl radical dimer under compression up to 14 GPa. Surprisingly, the isostructural radical dimer, where the hydrogen atom replaces fluorine, has remarkably different behavior under higher pressure despite the similarity in the structure and transport properties at ambient pressure. The dimer with hydrogen [1a]_2_ undergoes phase transformation at around 5 GPa, with good agreement with the experimental part of the work, whereas the fluorine dimer undergoes uniform compression without phase transition. The structural evolution and related changes in the molecular and band electronic structures of the two compounds were impressively replicated by DFT calculations. In the case of [1a]_2_, abrupt changes were observed in the lattice parameters and bandgap values as an effect of pressure. From enthalpy difference calculations, it was projected that, at low pressures, the uniformly compressed σ-dimer of the [1a]_2_ structure is more stable, with the buckled π-dimer taking dominance at higher pressures. The pressure transition was calculated to be in good accordance with the experimental results. The authors run two sets of calculations of [1a]_2_ dimers as geometry optimization was performed at 0 K and the spontaneous phase transition from the σ- to π-dimer was not expected due to the energy barrier. To overcome this issue, for the π-dimer form, the observed crystal geometry at different pressure values was used as an initial structure for calculations. The second set included isostructural compression of the ambient pressure crystal geometry of the ambient structure of [1a]_2_ to generate the equation of state for the hypothetical σ-dimer variant of [1a]_2_. For [1b]_2_, the crystal geometry at various pressure was applied as a starting point for geometry optimizations. 

In the last work of this section [47], the aim of the study was to investigate the high-pressure polymorphism of L-threonine within the pressure ranging from ambient pressure to 22 GPa using single-crystal X-ray and neutron powder diffraction. Due to the limited scattering geometry of the diamond anvil cell, the high-pressure study suffered from low completeness. To overcome this issue, the bond distance and angles were restrained to values observed at ambient pressure. The periodic DFT calculations were applied to validate the high-pressure structural models. Therefore, the geometry optimizations were performed on single-crystal structures at ambient pressure and those up to 22.31 GPa with cell dimensions and space group symmetry restraints. From a comparison of the bond distance and angle changes, it was detected that the torsion angles did vary significantly with pressure. Experimentally, it was noticed that the compound exhibited three phase transitions triggered by increasing pressure. DFT calculations were also used to calculate the pressure dependence of the lattice energy of L-threonine. The lattice energy increased gradually in phases I, I′, and II, and a discontinuity occurred during the II to III phase transition at 18.2 GPa, resulting in the abrupt destabilization of the lattice energy.

As stated above, CSP methods have found application also in DFT calculations at high pressure. Currently, there are multiple available codes that enable such predictions, i.e., GRACE, Polymorph Predictor, or USPEX. This last one has gained particular attention in works describing DFT calculations at high pressure. Universal Structure Predictor: Evolutionary Xtallography (USPEX) is an algorithm and software package developed for the prediction of crystal structures. It employs an evolutionary line of action in the investigation and prediction of the stable or metastable crystal structures of compounds. The algorithm’s ability to propose novel crystal structures has proven to be a great asset for material discovery and understanding structure–property relationships, and a great help in guiding experimental synthesis works.

In the first work [23], the authors used the USPEX algorithm to find the candidate high-pressure structure of ethylenediamine bisborane (EDAB); see Figure 6. As a result, they obtained 825 structures and, to shorten the list, they also compared the experimental and simulated XRD patterns of these structures. The 14 most promising structures for phase II (at 1.72 GPa) and 17 candidate structures for phase III (at 11.44 GPa) were subsequently optimized using the nonlocal van der Waals density functional theory (vdW-DF). This is regarded to give more accurate relative enthalpies than the semilocal GGA PBE functional. Following this, the calculated differences in enthalpies and parameters allowed the authors to establish the final top seven structures. Simultaneously, they performed calculations with the use of the PBE functional, which led to the same sequence of the lowest-enthalpy structures in the 0−17 GPa range, although with different values of transition pressure. However, DFT calculations suggest that phase II is *P*2_1_/c (A) and phase III is *P*2_1_/c (B), irrespective of the functional used. To confirm these identifications, Rietveld’s refinement method was employed to enhance the experimental XRD patterns for phases I−III using the corresponding theoretical crystal structures. Based on a comparison of the experimental and theoretical structures, the agreement for phase I was almost perfect. As compressed EDAB could be a potential candidate for hydrogen storage, its mechanical properties have been investigated at high pressure. The discontinuity of the cell parameters on the pressure reveals two phase transitions at around 1 and 7 GPa; however, the second drop is relatively small, suggesting the coexistence of phases II and III, which is supported by experimental results. In addition, the authors performed phonon calculations using solely the PBE functional for the PBE-optimized parameters since vdW-DF was not accessible in the version of Quantum Espresso used. The EDAB structures classified as phases I−III were confirmed to be true minima by the phonon calculation as no negative frequencies were detected.

In the next work [59], the authors used USPEX algorithms to generate candidate structures of croconic and squaric acid under 25 GPa. In order to establish the phase transition, the pressure dependence of the O-H bond under compression and decompression was examined using DFT calculations. In the case of croconic and squaric acid, isotropic compression predicts the formation of the crystalline structure at 12 GPa and 15 GPa, respectively. The formation of a new polymeric structure of croconic acid occurs at around 15 GPa under isotropic compression. In addition, the outcomes of evolutionary simulations at 25 GPa revealed that the structures of both acids are comparable to structures obtained under isotropic compression in terms of enthalpy. Upon decompression for both acids, higher pressure stabilizes the structure to lower pressure. It was noticed that the phase transition of squaric acid could be found under isotropic compression and USPEX evolutionary simulations and it was associated with the formation of O-H-O weak covalent bonds, and the high-pressure form had the high symmetry group *I*4 *m*. Based on the calculated dispersion phonon curves, the lack of negative (imaginary) frequencies indicates that structures obtained at 25 GPa are stable for both croconic and squaric acids. The authors also compared the simulated Raman spectra with the experimental data. 

Another example where the evolutionary structural search algorithm USPEX was applied to predict stable polymorphic structures at high pressure was described by Clarke et al. in the case of α-glycylglycine (α-digly) [46]. From the results of their calculations, the orthorhombic phase is suggested to be the lowest-enthalpy polymorph above 6.4 GPa. However, the discrepancy between the measured and simulated PXRD implies that the orthorhombic structure (digly-*P*2_1_2_1_2_1_) was not experimentally observed and the monoclinic phase persisted for the full pressure range. Both Raman and PXRD data indicated that the *P*2_1_/*c* symmetry was maintained, suggesting an isosymmetric phase transition at around 6–7.5 GPa. Hence, the transition to the orthorhombic phase requires a change in packing and bending peptide backbone; the transformation to this form might not have been noticed in the experiment due to the large energy barrier. An isosymmetric phase transition from α-digly to α′-digly was predicted at 10 GPa, as ZPE was not included in the enthalpy, or 11.4 GPa, when the ZPE was added, whereas the experimental transition pressure was ca. 6.7 GPa. However, it is worth mentioning that the differences in enthalpy in the 6–9 GPa range were smaller than 10 meV/atom. In this work, the thermal contribution to the energy was not included regarding the fact the supercell approach should be applied, which is very computationally expensive. Adding the ZPE is supposed to give results in better agreement with the experiment, with a simultaneously lower computational cost. Nevertheless, the changes in the a- and c-axes’ lengths under compression agreed with the experimental observations. The largest differences were observed for the b-axes. This could be associated with the fact that the b-direction includes multiple H-bonding interactions, which are known to be difficult to accurately model using the PBE functional. 

The USPEX algorithm was also used for the prediction of structures of 2,4,6-trinitro-3-bromoanisole (TNBA), an energetic material described in the previous section, “High-Energetic Organic Materials”.

### 4.2. Vibrational Spectra

Pressure-induced phase transition can not only affect structure parameters but also modify the vibrational spectra obtained for studied compounds. Therefore, the examination of vibrational modes by in silico methods could potentially bring great insights into the analysis and assignment of spectral features to different molecular motions, such as stretching, bending, or torsional vibrations, as the pressure changes.

Confirmation of this statement can be found in the work about glycinium maleate [40]; the DFT calculations can be used to assign inter- and intramolecular modes and help to describe local changes in molecules under compression. Based on theoretical research, the band evolution of glycinium maleate associated with the internal modes strongly indicated that the conformation was more correlated with the maleic acid molecules than with the glycine molecules. The DFT calculations were particularly useful in the assignment of the bands between 3050 and 3100 cm^−1^ due to extensive changes in the intensity of the bands under compression.

The authors of the next work [58] investigated the behavior of chloroform and dichloromethane upon compression and suggested the type of interaction that governed the compressibility. They found that for both compounds, the most stable phase was the ambient-pressure structures (*Pnma* and *Pbcn* for CHCl_3_ and CH_2_Cl_2_, respectively), which persisted up to 32 GPa, and the calculated phonon dispersion curves did not exhibit imaginary frequencies, suggesting the stability of these structures at high pressure. Furthermore, their results indicate that the P*6*_3_ structure of CHCl_3_ is metastable. Comparison of the calculated results with the experimental information of the Raman band positions allowed them to characterize the pressure-induced evolution of the crystal structures of both compounds.

In the next work [60], DFT calculations were used to simulate the Raman spectra of a single molecule and crystal of diisopropylammonium perchlorate (DIPAP) to assign and compare the calculated results with experimental measurements. In the case of molecular calculations, it was noticed that including the dispersion correction had no significant impact on the vibrational frequencies in the region 200−1650 cm^−1^. However, there was a slight shift in the higher wavenumber mode. For solid-state calculations, the results were in good agreement with experimentally obtained intermolecular and intramolecular vibrational modes as dispersion correction has a minor impact on the accuracy of theoretical results. 

In the next study [63], the authors investigated the pressure dependence of the Raman spectra of sorbic acid crystals to acquire insights into their vibrational properties. The observed broadening of the Raman bands due to the decrease in the cell volumes and higher disorder, the upshift of the wavenumbers, decreases in the Raman intensities, and some discontinuities were the main observed differences as an effect of the applied pressure. It is worth mentioning that at 0–6.0 GPa, which is in the range of expected transition pressure, the pressure dependence wavenumbers were nonlinear. The alterations detected in the Raman spectra’s external mode area were interpreted as changes in hydrogen bonding, resulting in changes in the conformation of the molecules. This interpretation was supported by the observation that the main changes observed among the internal modes were associated with the bands of a carboxylic group that actively took part in the hydrogen bonds. Based on the results, it was stated that the sorbic acid underwent a conformational transition due to the similarity of the Raman spectra between the room-temperature monoclinic structure and the high-pressure phase.

In the next work [54], the authors investigated the α, β phases of resorcinol by combining Raman, time-domain terahertz, and inelastic neutron scattering spectroscopy with solid-state density functional theory (DFT) calculations. The crystal structures of both polymorphs were optimized by using various GGA functionals to choose the most accurate by calculating the mean absolute errors between experimental and calculated internal coordinates. Based on the results, PBE TS was chosen for the examination of the lattice constants of both phases under compression. Even at high pressure, the deviations from the predicted values were less than 3%, except for the b-constant in the β phase, which was observed to decrease more easily. According to the vibrational spectra analysis, it was indicated that the standard (PBE) and “hard” (rPBE) GGA functionals were able to successfully reproduce the whole spectral range, aiding in the detailed interpretation of the bands. 

In the next study [36], the authors investigated the structural and vibrational properties of 1,3,5,7-cyclooctatetraene (COT) under pressure. The calculated Raman-active modes of a single COT molecule were in good agreement with experimental values. The theoretical lattice constants obtained from ab initio molecular dynamics simulation at 0 GPa and 3.8 GPa were in excellent accordance with measurements. 

### 4.3. Enthalpy (ΔH) Calculations 

In most cases, the transition pressure has been calculated based on differences in enthalpy between polymorphs, since the calculations are conducted at 0 K, where the enthalpy is equal to the Gibbs free energy. The phase transformation occurs at a pressure value where ΔH is equal to zero. As shown in the examples below, the predicted transition pressure from ΔH calculations can agree well with experimental findings [48,50]. However, there have been reported cases where a lack of entropy and thermal contribution could lead to the incorrect order of stability of the polymorphic phases, as in the case of glycine [94], or not even indicate the experimentally observed phase transition, as for the α and β phases of resorcinol [41].

In the first study [48], the authors performed dispersion-corrected DFT calculations of the α and β phases of L-threonine to investigate their structural, electronic, and dielectronic properties at hydrostatic compression. The root mean square deviation (RMSD) of the lattice constant between the calculated and experimental values was ca. 1% and the difference in intra- and intermolecular bond lengths and angles was around 6–8%, corresponding to the experimental results. From the differences in enthalpy (ΔH) values, the phase transition from the α and β phases was observed at 1.5 GPa, which agreed with the experimental work. On the pressure dependence of lattice parameters, a discontinuity at 1.5 GPa was noticed, suggesting the occurrence of a phase transformation. The analysis of changes in the lattice parameters of intra- and intermolecular bonds revealed that the changes in lattice structure played a predominant role in the pressure effect on intermolecular hydrogen bonds, but not the change in intramolecular bond lengths. Furthermore, it was established that the β form could exist without pressure as a metastable phase. In the next work [57], Howard et al. studied the phase transition and thermal expansion of ammonium carbamate. This compound is known to exist in two polymorphic forms, namely α and β. By employing DFT calculations, the authors showed that the thermodynamically stable phase at ambient pressure is the α-polymorph, with a calculated enthalpy difference with respect to the β-polymorph of 0.399 kJ mol^−1^. According to the calculation results, the transition to the β-polymorph could occur at 0.4 GPa. However, this observation has not yet been studied using experimental methods. Another interesting aspect of this work was that the authors considered both the dispersion-corrected (TS) and non-corrected DFT calculations. For the work reported, they found that geometry optimizations completed without dispersion corrections led to disagreements of 10% in the unit cell parameters, these being reduced to 1–3% with the use of dispersion corrections. Similarly, the estimated phase transition pressure reported later was reduced from 2.0 to 0.4 GPa. In contrast to other similar works reported, when determining the pressure of phase transition, the authors compared solely the ΔH values instead of ΔG. This was due to the calculations being performed solely at 0 K, without taking into account either ZPVE or entropy changes. 

In another work [55], the authors used DFT methods to investigate the structural changes of glycine and L-alanine crystals under pressure up to 10 GPa. They performed calculations with and without van der Waals interaction correction. Based on the results, the authors established that the inclusion of van der Waals interactions is crucial for an accurate description of the intermolecular interactions of amino acids. The differences in the stability of glycine polymorphs are very small, so it is challenging to accurately establish the stability order. Furthermore, the calculated differences are sensitive to the exchange–correlation functionals applied. In the case of glycine, the calculations with the vDW interaction better predicted the stability order of polymorphs, unlike calculations without this component. Based on the calculated enthalpy values, the pressure-induced phase transition of the β to δ phase was predicted at 4.0 GPa, 1.5 GPa, and 0.7 GPa for PBE, vdW-DF, and vdW-DF-c09x, respectively, whereas, experimentally, it occurs at 0.76 GPa. Consequently, it was stated that PBE without including dispersion correction overestimated the transition point. For L-alanine, only the vDW-aware functionals correctly determined the unit cell parameters under pressure. 

In the next work [56], the DFT and PIXEL methods were used to research the behavior of L-serine at pressure up to 8.1 GPa. The experimental structure parameters were reoptimized with fixed unit cell dimensions and unrestricted other parameters and symmetry. The obtained geometries were in good agreement with the neutron powder diffraction analysis for each phase, and the root mean square deviation (RMSD) of the bond length in the structures was never greater than 0.08 Å. Based on the results, performing the geometry relaxation of an ambient-pressure form with a fixed external pressure parameter could be a powerful tool for the prediction of the high-pressure-induced differences in molecular packing and geometry. The PIXEL method is based on the determination of a molecular electron density map, followed by processing the map into larger pixels and then the calculation of energy terms between pairs of pixels in adjacent molecules. The technique does not take into account any intramolecular changes in energy and it is used to calculate the energies of intermolecular interactions. However, the transition from phase I to phase II is partly driven by a conformational change in the molecule. According to PIXEL calculations, there is a significant energy gap between the intermolecular energies of phases I and II. The phase transition from L-serine I to L-serine II occurs near 5 GPa, where there is a break in the gradient of the curve, with the enthalpy of phase II being more negative after the change than the extrapolated values for phase I due to the stabilization of the internal energy of the serine molecules and volume reduction. It seems that the II to III phase transformation is driven by rearrangement in intermolecular interactions.

In a subsequent paper examining the behavior of L-serine crystals at high pressure [50], the authors applied DFT calculations to investigate the mechanism of reversible phase transition between polymorphs of small organic molecules like L-serine. Firstly, they performed optimizations of the crystal structures documented to dominate at these pressures. However, this approach did not allow the researchers to compare the enthalpies and crystal energies of different phases at the same pressure and explain the occurrence of phase transition. Hence, the authors performed optimization of all polymorphs at the same pressure range, outside of the range of their existence. Based on the enthalpy differences (ΔH) between polymorphs, the pressure transitions were defined as 3.7 GPa and 5.3 GPa for the transition from phase I to phase II and phase II to phase III, respectively. Although the calculated values were smaller than the experimentally registered 5.3 GPa and 7.8 GPa, respectively, the difference between the points of the two-phase transitions was close to that observed experimentally. The discrepancy might occur due to the limitations of the model or might suggest kinetic control over the transition. Additionally, the polymorphs of L-serine have been experimentally observed to coexist in a wide pressure range; thus, the transitions could be initiated at different pressure values, depending on the experimental conditions. Although the entropy factor was not included, the computational simulations correctly predicted the order of the polymorphs’ stability. As, during phase transitions I–III, the crystal symmetry did not change significantly and no crucial changes were observed in the Raman spectra, the authors assumed the differences in the TΔS term into Gibbs free energy to be small and not including this factor allowed them to reduce the computational time, with minor inaccuracy for the Gibbs energy. Based on the research, the transition of L-serine polymorphs was shown to be triggered by the PV term, at least between the I and II phase. The investigation of the effect of pressure on H-bond interactions in gas-phase cluster models helped to highlight the significant difference between the I→II and II→III phase transitions. The first transition is triggered by the large overstrain of a selected intermolecular hydrogen bond, experimentally manifested in changes in cell parameters, and the large hysteresis; the latter is proposed to be accompanied by multiple small changes in various hydrogen bonds. The current study demonstrates that relatively simple calculations with the combination of detailed experimental data may provide insight into the macro- and micro-driving forces of pressure-induced phase transition in hydrogen-bonded molecular crystals. 

### 4.4. Gibbs Free Energy (ΔG) Calculations

In order to obtain the thermodynamic properties, the very computationally demanding and time-consuming phonon density of states calculations need to be performed. Therefore, the size of the investigated crystals and the required computer parameters could limit the application of this approach to determine the relative stability and transition pressure. Furthermore, the calculation of thermodynamic functions can help to establish whether the phase transition is enthalpy- (e.g., glycine [49]) or entropy-driven (e.g., chlorothiazide [39], Figure 7).

The study of glycine [49] is one example where the Gibbs free energy (ΔG) was calculated to predict phase transformation. The aim of the work was to evaluate whether periodic DFT calculations could be applied to the investigation of the dependence of increasing pressure on the molecular crystal structures of glycine and their stability. Firstly, the authors tested various functionals with or without dispersion correction to choose the most accurate based on the results of the geometry optimization of the γ and δ phases at 3.27 GPa, as, at this pressure value, both forms have been observed experimentally. According to the outcomes, the GGA functionals provided more accurate results than LDA. Surprisingly, including OBS dispersion correction decreased the accuracy of the LDA calculations. For further investigation, the PBESOL functional was chosen, since it was found to be very accurate in most cases, but also was originally developed for densely packed solids. The next step was to perform geometry optimization up to 7.8 GPa for both polymorphs. The obtained results of the lattice parameters were found to be in very good agreement with the corresponding experimental data. To test the accuracy of the calculations, the PBE functional without dispersion correction at pressure values of 0 GPa, 5.83 GPa, and 7.80 GPa was applied. From the comparison of the results obtained using two different functionals, the crystal structures were significantly less accurately modeled using PBE than PBESOL. Additionally, the authors suggested that the simulated lattice parameters in some cases could be even more accurate than experimentally measured as they better fit the overall trend of changes in lattice parameters within increasing pressure. Performing thermodynamic parameter calculations of the γ and δ phases at various values of pressure allowed them to establish the order of stability and the transition pressure at which the order was reversed. The calculated values of Gibbs free energy (ΔG) were in good accordance with the experimental findings, where the γ phase started to transform into the δ polymorph at ca. 2.74 GPa. Furthermore, based on the computational results, the phase transition of glycine was described as enthalpy-driven. Nonetheless, the entropy effect was also favorable for the transformation to occur at high pressure.

### 4.5. Phonon Calculations

Since, in some cases, the predicted high-pressure structures could be metastable phases, to test their stability, the phonon frequency can be calculated to detect the presence of potential imaginary frequencies [23,58,59,61]. The lack of negative frequencies suggests that the forms predicted by theoretical methods are stable under the investigated conditions. For instance, no imaginary frequencies were observed for the structures of croconic and squaric acid obtained at 25 GPa for both acids [59], or in the case of chloroform and dichloromethane for the structures obtained at 32 GPa. However, the previously proposed structure *P*6_3_ of CHCl_3_ was found to be a metastable form; thus, the ambient-pressure structure is likely to be a ground-state polymorph of this compound up to 32 GPa [58]. Comparably, based on phonon calculations, the structures of EDAB identified as the I–III phases were proven to be true minima [23]. The phonon dispersion curves were evaluated along the high-symmetry direction to obtain information about the structural stability of the g, α, d, and β phases of C_11_N_4_. According to the results, the g phase was the most energetically stable at ambient pressure, and α- and d-C_11_N_4_ indicated no negative frequencies, in contrast to β- and g-C_11_N_4_ [61].

## 5. Other Aspects Associated with DFT Calculations at High Pressure

In this section, the problems and phenomena typical of DFT studies at high pressure are presented. These aspects include methods of determination of the pressure-induced phase transition conditions, energy barrier value calculations, situations in which no pressure-induced transformations are observed, or applications of computational anisotropic compression.

### 5.1. Determination of Pressure-Induced Phase Transition Conditions

There are several known methods to determine the transition pressure based on structural, energetic, or property changes. Below, some of them are listed with a short description of the work, to give an overview of the methods used and where they have been applied. Notably, the mentioned approaches are not limited to the given examples, and some of them, solely or in combination with different approaches, have been used in other papers. 

#### 5.1.1. Common Tangent to the Two E(V) Curves, *p* = −dE/dV

As shown in the case of C_11_N_4_ study [61], the transition pressure can be obtained directly from the calculated energy–volume curves. The formation enthalpy under the pressure is defined as H = E (V) + PV, where E is the total energy for the cell with volume (V). The transition pressure for each phase transformation was obtained by calculating the common tangent slope of the two energy–volume curves. It was shown that the transition from g-C_11_N_4_ to α-C_11_N_4_, d-C_11_N_4_, and β-C_11_N_4_ can occur at 3.557 GPa, 9.468 GPa, and 46.032 GPa, respectively. The obtained data indicated that the transition from g-C_11_N_4_ to α-C_11_N_4_ and d-C_11_N_4_ occurs at low pressure, and the transition from g-C_11_N_4_ to β-C_11_N_4_ occurs in the high-pressure phase. Additionally, the authors investigated the phase transitions and vibrational, mechanical, and thermodynamic properties of four polymorphs. Thanks to the performed calculations, the prediction of the mechanical and thermodynamic properties, including the bulk modulus, heat capacity, and thermal expansion coefficient, of C_11_N_4_ polymorphs was possible.

#### 5.1.2. Changes in Properties Observed upon Compression

In some papers, the possibility of pressure-induced phase transition was suggested based on changes in properties, e.g., upon noticing alterations in bandgap widths for ε-CL-20, 400 GPa was supposed to be the critical pressure for insulator–metal transition [37], or in the case of the previously described LLM-105, the abrupt bandgap decrease suggested a pressure-induced phase transformation, confirmed by experimental work [24]. The electronic properties can change with an increase in pressure, which can be used in tuning materials’ properties by exposing them to a high pressure factor [113]. Consequently, the theoretical investigation of electronic and mechanical properties can be a beneficial tool in designing materials with desired properties, e.g., for photovoltaic applications. 

In one work [64], single-point DFT band structure calculations were performed to investigate the pressure-induced changes in the electronic structures of the α, β, and γ phases of the oxobenzene-bridged 1,2,3-bisdithiazolyl radical conductor (3a) obtained at 0 GPa (α phase), 6.0 GPa (β phase), and 11.1 GPa (γ phase). The α phase is a Mott insulator, as the bandwidth of all explored bands was small. The calculated results showed that the increase in pressure resulted in an increase in bandwidth in the β phase; further compression initiated the second phase transition to the γ phase, lifting the HOMO band and causing higher-lying virtual orbitals to drop. Generally, the presence of low-lying LUMO triggers high electron affinity and creates an electronically much softer radical with a low on-site Coulomb potential U, providing an important insight into the design of radical-based conductors. 

In the study of the polymer [Zn(μ-Cl)_2_(3,5-dichloropyridine)_2_]_n_ [13], as was mentioned before, the transition associated with breaking symmetry from $P\bar{4}b2$ to $P\bar{4}$ was predicted by using DFT calculations based on the pressure-induced softening of low-frequency vibrations in the Raman spectra [13]. Therefore, calculations can support the interpretation of experimental spectra and help to detect structural phase transitions, although the pressure–volume (p-V) curves may not show any discontinuities.

### 5.2. Lack of Pressure-Induced Phase Transition

Although phase transition has not been observed in all studies, DFT calculations have allowed us to simulate the effect of high pressure on the structural, mechanical, and electronic properties of investigated materials. The lack of structural phase transition was observed, e.g., in the study of TATB [28], LLM-105 [31,32], and silver fulminate [34] or aspirin and paracetamol, as described below [53]. 

In one study [53], the authors investigated aspirin and paracetamol’s (Forms I and II) behavior up to 5 GPa using long-range dispersion-corrected hybrid density functional calculations. In both cases, the phase transition was not observed, which was in line with the experimental results of paracetamol, and no investigational results for aspirin were available to compare at that moment. In another investigation, it was noted that the phase transition between Form I and Form II of aspirin was not detected up to 10 GPa using micro-Raman spectroscopy but, instead, Form I was transformed into a new phase, Form III, above ~2 GPa, and the authors proposed Form III to be the most stable polymorph of aspirin at high pressure.

Based on the pressure dependence of the lattice parameters and bond length of both polymorphs of paracetamol, the maximum compression was observed in the b-lattice direction, perpendicular to the hydrogen-bonded molecular layers. In the case of Form I, unusual behavior was observed as compression resulted in the expansion of the a-lattice parameter. Firstly, the a-edge decreased from 12.621 Å to 12.471 Å at 1.5 GPa and then increased up to 12.738 Å at 4.0 GPa. Although this is not ordinary behavior under compression, the expansion of the a-lattice parameter of Form I was experimentally observed at a pressure range of 2–4 GPa. 

From the research on aspirin, both forms were characterized by similar anisotropic strain at applied pressure. Furthermore, the IR spectra was calculated for each form at 0 GPa and 5 GPa as changes in intermolecular interactions manifested in the shifting of the bands. The pressure-induced structural alterations were complemented by bands with red shifts in the IR spectra of the four investigated forms.

The behavior of aspirin crystals under pressure was also an area of interest in a previously mentioned work [52], where the energy barrier was calculated, indicating that no pressure-induced phase transition was observed. 

### 5.3. Anisotropic Compression

Anisotropic compression can be simulated by using DFT calculations. It is especially beneficial in the investigation of energetic materials, as detonation is a non-equilibrium, ultrafast process with strong orientation dependence, and real-time measurements of this type of material are challenging. The pressure-dependent vibrational frequency shifts and the possibility of phase transformation under various compression orientations could remain elusive [25]. Therefore, a theoretical approach can support the examination and give valuable insights into the behavior of materials under non-hydrostatic pressure, as in the study of energetic materials FOX-7 [25], β-HMX [33], and MAPbBr_3_ [17] or polymer Zn(μ-Cl)_2_(3,5-dichloropyridine)_2_]_n_ [13] and as described below for oxalic acid [62].

In their work [62], the authors analyzed dihydrate and the α and β polymorphic forms of anhydrous oxalic acid. The calculated structural parameters were in good agreement with the experimental data. The sets of geometry optimization were performed at several different pressure values applied along the direction of the minimum Poisson ratio. According to the obtained results, it was shown that the dihydrated form of oxalic acid undergoes a phase transition as an effect of negative pressure induction smaller than approximately −0.045 GPa. Both forms of anhydrous oxalic acid also undergo pressure-induced phase transitions, in this case for positive pressures, larger than around 1.91 GPa for the α polymorph and around 0.21 GPa for the β polymorph. It was noticed that α- oxalic acid underwent a pressure-induced phase transition under the effect of applied pressure directed along the (0.01, 0.73, 0.68) direction, whereas for the β phase, this occurred along the (0.95, 0.00, 0.31) direction.

### 5.4. Polymorphic Transition Energy Barrier Calculations 

In the case of indole, as well as other compounds, it has been shown that, sometimes, geometry optimization, even at a higher level of theory, is not sufficient to observe phase transition. In one work [66], the authors investigated indole crystals’ responses to hydrostatic pressure using molecular dynamics and DFT calculations. Both methods effectively reproduced the experimental structure with an error of around 1%. However, abrupt changes in lattice parameters were not observed in the DFT calculation results, even though the HB to β phase transition energy was relatively small (around 0.1 eV per unit cell). Nevertheless, the transition barrier was ca. 0.9 eV per unit cell and required increasing kinetic energy, which was not provided in the static calculations performed at 0 K. These results revealed that the pressurization was not sufficient for the transformation to occur.

## 6. Conclusions

Density functional theory is a potent and often used quantum mechanical tool for the examination of different features of matter. The studies in this area cover a wide variety of topics, including the creation of original analytical methods centered on the creation of exact exchange-correlation functionals and the application of this method to the prediction of the molecular and electronic configurations of atoms, molecules, and solids in both gas and solution phases. Since there are still problems to be solved, designing and evolving more effective density functionals is a continual process. Ensuring the appropriate qualities at a reasonable processing cost is a major quantum task. Future work will concentrate on creating even more consistently accurate density functionals for particular applications, enabling researchers to utilize DFT’s relatively high precision at a low cost.

The creation of precise force fields generated from first-principles data is perhaps one of the most significant directions in the first-principles modeling of molecular crystals. The types of molecular dynamics simulations and multiscale models required to properly comprehend the thermodynamics and kinetics of molecular crystals as a function of temperature and other factors would be made possible by such force fields. A complete picture of the creation, stability, and characteristics of molecular crystals may be obtained by properly determining the underlying electronic energies and significant response qualities using entirely first-principles approaches.

The above review shows that DFT calculations can be successfully used to describe the various molecular solids, including organometallic compounds and high-energetic materials, which are particularly demanding and hazardous from the experimentalist’s point of view. Moreover, the types of calculations and modeled properties are diverse and include not only the most popular geometry optimization but also the ab initio molecular dynamics, spectral (Raman, UV–Vis, IR), or thermodynamic property calculations. Therefore, the application of DFT methods can lead to a better understating of the changes in the structures and properties of materials resulting from the application of high pressure.

Various strategies are being used to determine whether the pressure-induced polymorphic transition should occur, and, if so, at what conditions. The least computationally demanding is the analysis of a common tangent to the two E(V) curves, as it does not require the phonon density of states calculations. Other methods include the calculation and comparison of Gibbs free energy or molecular dynamics simulations.

While, in most of the works, isotropic compression has been used to reproduce the commonly applied experimental conditions, in the reviewed works, successful applications of anisotropic compression have also been found.

Finally, it is worth noticing how different levels of theory imply the accuracy of the results. As in other areas of molecular modeling, in the DFT calculations of organic solids under high pressure, more computationally demanding calculations and more complex systems usually yield more reliable results, with the cost being computational time. For example, while the first-principles methods enable us to calculate lattice energies with accuracy better than 1 kcal/mol, the requirement is to go beyond a pairwise model of dispersion correction and include non-additive many-body contributions—for example, by using the MBD method. However, this approach is significantly more computationally demanding than the use of the TS or Grimme schemes. Moreover, as described above in the examples of urea and chlorothiazide, sometimes, to observe the polymorphic phase transition, molecular dynamics simulations are required, as geometry optimization may not be sufficient due to the entropy-driven transformations. However, the computational cost of MD simulations is much higher than that of geometry optimization. Another example is the size of the cell being optimized, as sometimes the modeling of a supercell composed of a few (or more) unit cells provides more accurate results, especially when the crystal symmetry is changed during optimization. Unfortunately, an increase in the size of the modeled system leads to the undesired elongation of calculations.

The most popular programs that are used to perform such calculations are plane-wave DFT codes, such as CASTEP, VASP, or Quantum ESPRESSO (Table 2). Fortunately, most of them are free to use for either non-commercial or academic purposes. Moreover, due to the variety of published tutorials in the form of both videos and pdf documents, it is feasible to start performing such calculations even without the use of introductory courses.

Therefore, in conclusion, we would like to encourage researchers who are yet not familiar with such methods to consider their use, as they can be extremely helpful both in planning experiments and during the analysis and interpretation of experimental results (Figure 8).

## Figures and Tables

**Figure 1 ijms-24-14155-f001:**
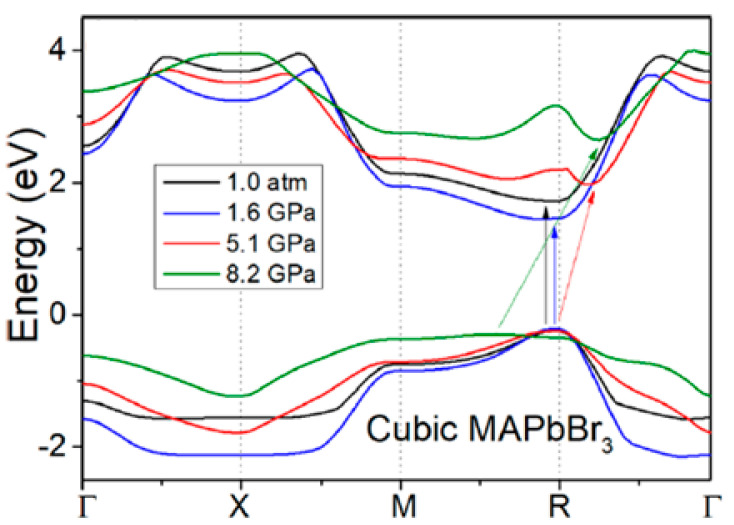
Band structures close to the Fermi level of cubic MAPbBr_3_. Adapted with permission from [16]. Copyright 2023 American Chemical Society.

**Figure 2 ijms-24-14155-f002:**
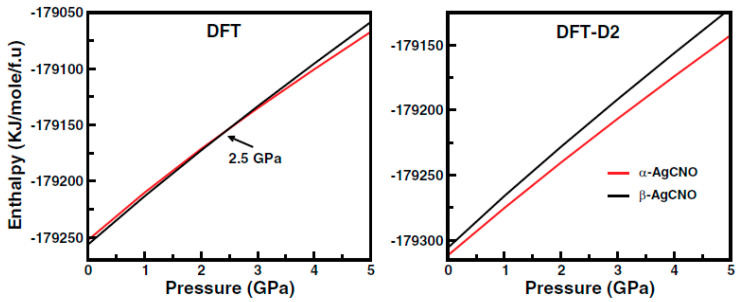
Calculated enthalpy as a function of pressure for α- and β-polymorphic phases of AgCNO with (DFT-D2) and without (PBE-GGA) dispersion correction method. Reprinted from [34], with the permission of AIP Publishing.

**Figure 3 ijms-24-14155-f003:**
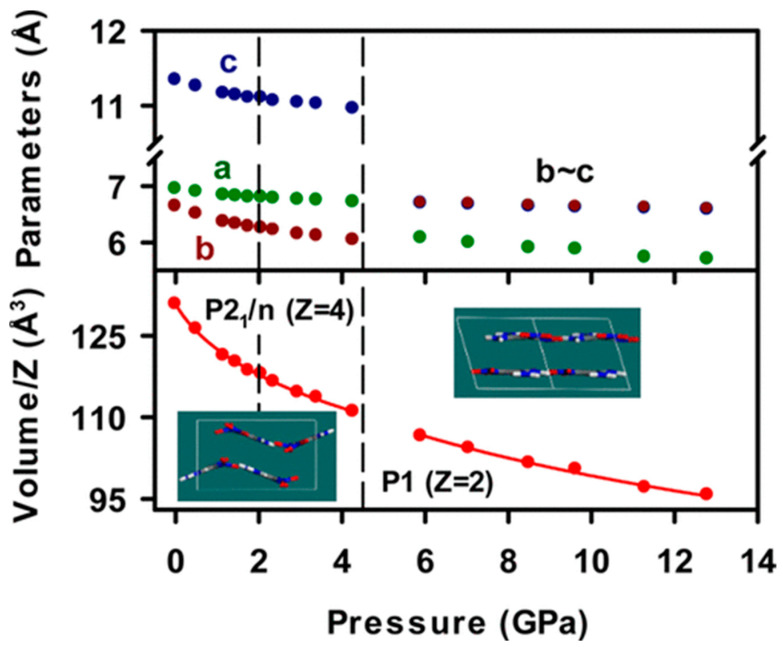
High-pressure structural response of an insensitive energetic crystal—1,1-diamino-2,2-dinitroethene (FOX-7); a, b, c—unit cell lengths. Adapted with permission from [29]. Copyright 2023 American Chemical Society.

**Figure 4 ijms-24-14155-f004:**
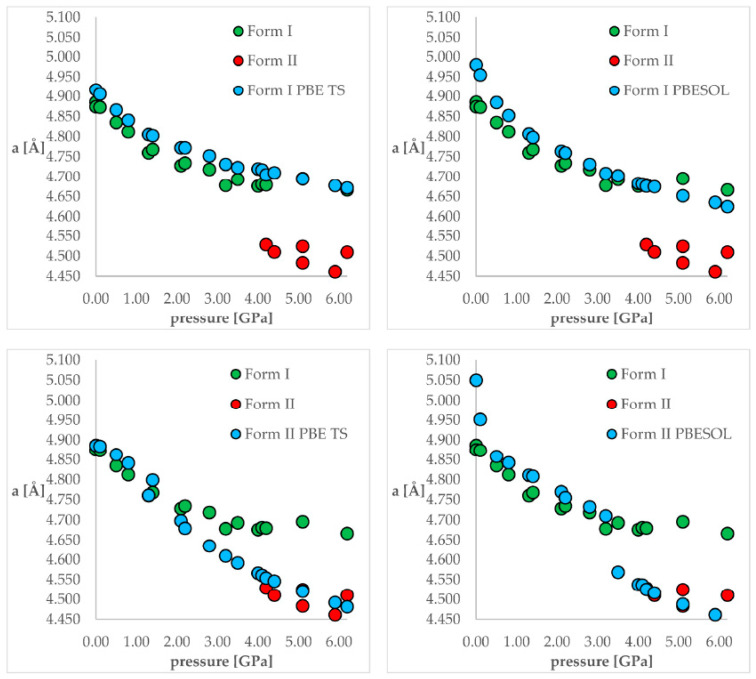
Geometry optimization of two polymorphic forms (I and II) of chlorothiazide at various pressure using two DFT functionals (PBESOL and PBE + TS). Adapted from [39], licensed under CC BY 4.0.

**Figure 5 ijms-24-14155-f005:**
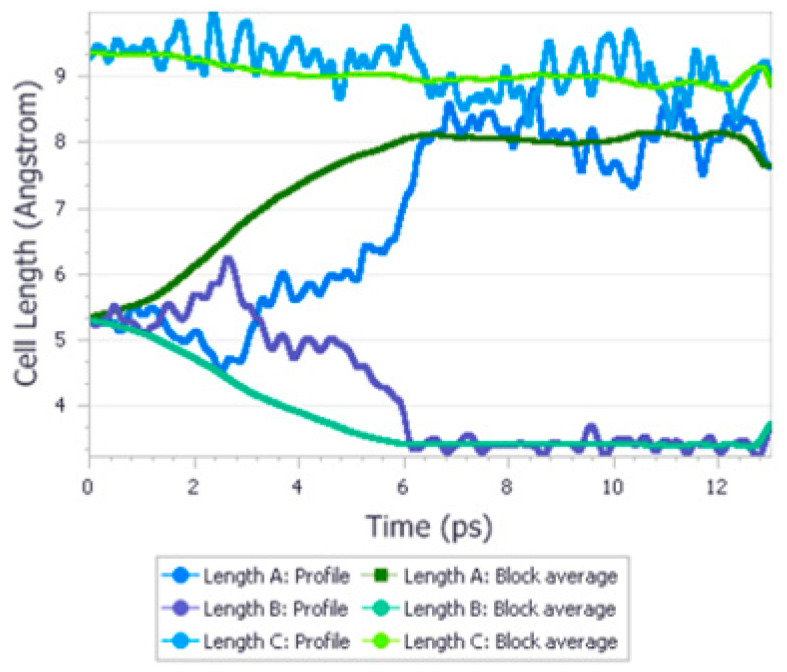
Ab initio molecular dynamics simulations of urea Form I at 3.10 GPa. Polymorphic phase transition is observed after 6 ps of simulation. Adapted from [45], licensed under CC BY 4.0.

**Figure 6 ijms-24-14155-f006:**
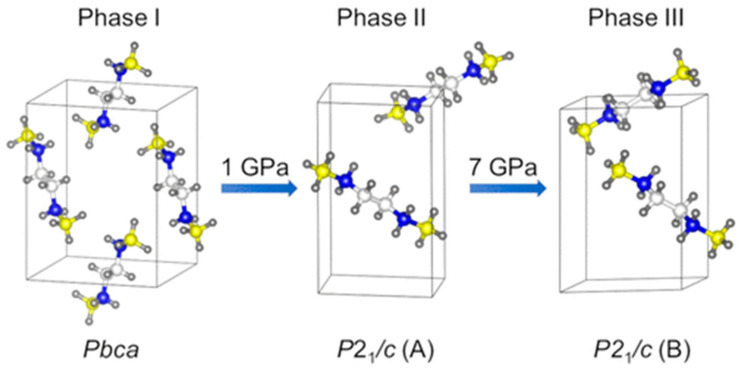
Pressure-induced phase transitions of crystalline ethylenediamine bisborane [23]. Copyright 2023 American Chemical Society.

**Figure 7 ijms-24-14155-f007:**
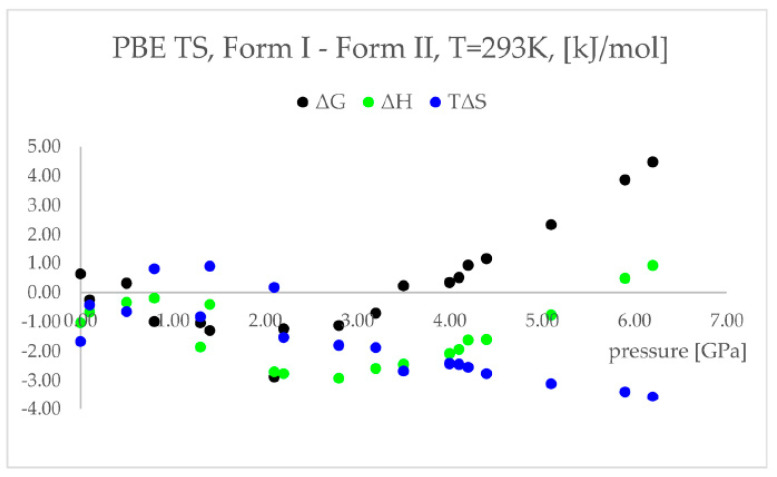
Thermodynamic calculations for chlorothiazide, indicating entropy-driven phase transition of Form I to Form II at higher pressure. Adapted from [39], licensed under CC BY 4.0.

**Figure 8 ijms-24-14155-f008:**
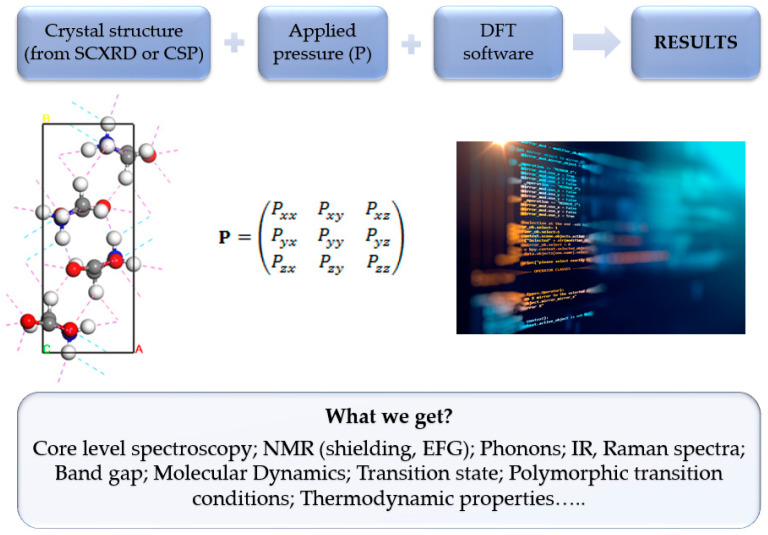
A summary of the requirements and possible outcomes of the DFT calculations of molecular solids under high pressure.

**Table 1 ijms-24-14155-t001:** Selected articles on the application of density functional theory (DFT) quantum mechanical calculations to study the high-pressure polymorphs of organic crystalline materials.

N°	Molecule	Polymorphs Studied	Pressure Range	Type of Calculation	Software Applied	Methods—DFT Functional and Dispersion Correction	Year	Ref. in Article
**A. Organic materials with metal additives**								
**1.**	(IPy)_4_(In_2_Cl_10_) IPy = 4-iodopyridinium	NP	0–1.51 GPa	GO, HS, NBO, DOS, OP, MO	Crystal Explorer; Gaussian	B3LYP	2021	[12]
**2.**	[Zn(μ-Cl)2(3,5-dichloropyridine)_2_]_n_	$P\bar{4}b2$, $P\bar{4}$	0–9.34 GPa	GO (compression and decompression), Raman	CASTEP	PBE TS	2021	[13]
**3.**	Methylammonium lead bromide (MAPbBr_3_)	$Pm\bar{3}m$, *Im*$\bar{3}$	0–2.5 GPa	GO, BG	Quantum ESPRESSO	PBE-D3 Grimme	2020	[14]
**4.**	CH_3_NH_3_PbI_3_ (MAPbI_3_)	Tetragonal, orthorhombic, and cubic structures	0–2 GPa	BG, DOS	VASP	PBE	2019	[15]
**5.**	Methylammonium lead bromide (MAPbBr_3_)	$Pm\bar{3}m$, *R3m*, *R3*	0–130 GPa	GO, BG	VASP	PBE	2019	[16]
**6.**	Methylammonium lead bromide (MAPbBr_3_)	*I* ($Pm\bar{3}m$*) II (Im*$\bar{3}$*)*, *III (Im*$\bar{3}$*)*, *IV (Pnma)*	0–3 GPa	GO, BG, aiMD	VASP	PBE	2017	[17]
**7.**	Methylammonium lead iodide (MAPbI_3_)	*I*4*/mcm*, *Immm*, *Im*$\bar{3}$	0–1.95 GPa	GO, BG	NP	PBE-D3, B3PW91 + SOC_PBE_	2016	[18]
**8.**	Pt(bpy)Cl2, bpy = 2,2′-bipyridine	Yellow and red form	0–3.8 GPa	GO, MO, TD-DFT	Gaussian	B3LYP, LDA, BLYP	2007	[19]
**B. High-energetic organic materials**								
**9.**	2,4,6-Trinitro-3-bromoanisole (TNBA)	*P*2_1_*/c*, *P*2_1_2_1_2_1_	0–10 GPa	USPEX, GO, BG, DOS	VASP	PBE-D2	2022	[20]
**10.**	Pentazolate anion (cyclo-N_5_^−^) salt 3,9-diamino-6,7-dihydro-5H-bis([1,2,4]triazolo)[4,3-e:30,40-g][1,2,4,5]tetrazepine-2,10-diium ((N_5_^−^)_2_DABTT_2_^+^)	NP	0–50 GPa	DOS, BG, PC, IR	CASTEP	PBE/G06	2022	[21]
**11.**	Pentazolate anion (cyclo-N_5_^−^) salt N-carbamoylguanidinium (N_5_^−^GU^+^)	NP	0–50 GPa	DOS, BG, PC, IR	CASTEP	PBE/G06	2022	[21]
**12.**	1,3,5-Trinitrohexahydro-s-triazine (RDX)	α, β, ε′	0–20.7 GPa	GO	VASP	PBE vdW correction	2021	[22]
**13.**	Ethylenediamine bisborane (EDAB)	I, II, III	0–17 GPa	USPEX, XRD, GO, PC, PF	DIAMOND; Quantum ESPRESSO	vdW-DF, PBE	2021	[23]
**14.**	2,6-Diamino-3,5-dinitropyrazine-1-oxide (LLM-105)	NP	0–25.7 GPa	GO, DOS, BG AS, ES	NP; Gaussian	PBE; B3LYP	2020	[24]
**15.**	1,1-Diamino-2,2-dinitroethene (FOX-7)	α, α′, β, γ, δ, and ε	0–30 GPa	GO, PC, Raman	CASTEP	PBE	2019	[25]
**16.**	2,4,6-Trinitrotoluene (TNT)	m-TNT and o-TNT	0–5 GPa	GO	CASTEP	PBE-D2 PBE TS	2019	[26]
**17.**	Octahydro-1,3,5,7-tetranitro-1,3,5,7-tetrazocine (HMX)	α, β	0–50 GPa	GO, HS AIM, IGM. MPD, MP, PC, PF, IR, DOS	CASTEP	PBE	2019	[27]
**18.**	Triaminotrinitrobenzene (TATB)	NP	0–27 GPa	GO, ZPE, Raman	VASP	PBE-D2 Grimme	2017	[28]
**19.**	1,1-Diamino-2,2-dinitroethene (FOX-7)	α, α′, ε	0–12.8 GPa	GO	CASTEP	PBE Grimme	2016	[29]
**20.**	Cyclotrimethylenetrinitramine (RDX)	α, γ	0–10 GPa	GO, TD, PC	CP2K	PBE-D3(BJ)	2016	[30]
**21.**	2,6-Diamino-3,5-dinitropyrazine-1-oxide (LLM-105)	NP	0–20 GPa	GO, aiMD	CP2K	PBE-D2	2015	[31]
**22.**	2,6-Diamino-3,5-dinitropyrazine-1-oxide (LLM-105)	NP	0–45 GPa	GO, MO, aiMD (at 0 Gpa)	CASTEP; CP2K	PBE-D2	2014	[32]
**23.**	Octahydro-1,3,5,7-tetranitro-1,3,5,7-tetrazocine (HMX)	Β-HMX, insulator, metal	0–130 GPa	BG, QMD+ MSST	CP2K	SCC-DFTB	2014	[33]
**24.**	Silver fulminate (AgCNO)	α, β	0–5 GPa	GO, PC, TD, BG, DOS	CASTEP; WIEN2k	PWSCF; PBE, PBE-D2; TB-mBJ	2014	[34]
**25.**	2,4,6-Trinitro-1,3,5-benzenetriamine (TATB)	NP	0–7.02 GPa	GO, CM	Quantum ESPRESSO	PBE, PBE Grimme	2010	[35]
**26.**	Cyclotrimethylenetrinitramine (RDX)	α, γ	0–3.36 GPa and 3.9–7.99 GPa	GO, CM	Quantum ESPRESSO	PBE, PBE Grimme	2010	[35]
**27.**	Hexanitrohexaazaisowurtzitane (CL20, HNIW)	β, γ, ε	0–2.7 GPa	GO, CM	Quantum ESPRESSO	PBE, PBE Grimme	2010	[35]
**28.**	Nitromethane (NM)	NP	0–7.6 GPs	GO, CM	Quantum ESPRESSO	PBE, PBE Grimme	2010	[35]
**29.**	Octahydro-1,3,5,7-tetranitro-1,3,5,7-tetrazocine (HMX)	α, β, δ	0–7.47 GPa	GO, CM	Quantum ESPRESSO	PBE, PBE Grimme	2010	[35]
**30.**	Pentaerythritol tetranitrate (PETN)	NP	0–9.04 GPa	GO, CM	Quantum ESPRESSO	PBE, PBE Grimme	2010	[35]
**31.**	1,3,5,7-Cyclooctatetraene (COT)	NP	0 and 3.8 GPa	GO, PC, Raman, XRD, aiMD	DMol3; Gaussian; CPMD	PW91	2008	[36]
**32.**	Hexanitrohexaazaisowurtzitane (CL-20, HNIW)	α· H2O, β, γ, and ε	0–400 GPa	GO, SP, DOS, BG	CASTEP; DMol3	PBE; rPBE	2007	[37]
**33.**	Triclabendazole	I, II	0–10 GPa	Supercell approach combined with the embedded fragment method, GO, PC, IR, Raman, TD	Gaussian	ωB97XD	2022	[38]
**C. Pharmaceuticals**								
**34.**	Chlorothiazide	I, II	0–6.2 GPa	GO, PC, TD, aiMD	CASTEP	PBE TS, PBESOL	2021	[39]
**35.**	Glycinium maleate	NP	0–5.6 Gpa	GO, PC, Raman	Quantum ESPRESSO	LDA	2021	[40]
**36.**	Resorcinol	α, β	0–4 GPa	GO, DFT, and DFTB3-D3(BJ) approach: vibrational frequencies	Quantum ESPRESSO; Phonopy; DFTB+	B86bPBE-XDM	2021	[41]
**37.**	Glycine	α, β, γ	0–50 GPa	GO, TD, BG	Quantum ESPRESSO	PBE-D3	2020	[42]
**38.**	L-Histidine	I, I′, II, II′	0–7 GPa	GO	CASTEP	PBE TS	2020	[43,44]
**39.**	Urea	Form I and IV	0 and 3.1 GPa	GO, PC, TD, aiMD	CASTEP	PBE TS, PBESOL, WC	2020	[45]
**40**	A-Glycylglycine	α, α′, *P*2_1_2_1_2_1_	0–18GPa	USPEX, GO, ZPE, PXRD	VASP, VASP VTST tools for ZPE	PBE-D2	2020	[46]
**41.**	L-Threonine	I, I′, II, III	0–22.31 GPa	GO	CASTEP	PBE	2019	[47]
**42.**	L-Threonine	α, β	0–5 GPa	GO	CRYSTAL	PBE-D3(BJ)	2019	[48]
**43.**	Glycine	γ, δ	0–7.8 GPa	GO, PC, TD	CASTEP	PBE, PBE Grimme, PBE TS, PBESOL, PW91, PW91 OBS, RPBE, WC, CA-PZ, CA-PZ OBS	2018	[49]
**44.**	L-Serine	I, II, III	0–8.2 GPa	GO (compression and decompression)	VASP; Gaussian	PBE-D; M06-2X	2017	[50]
**45.**	Tolazamide	I, II	0–20 GPa	GO, ZPE, TB	Gaussian; VASP	M062X; PBE-D3(BJ)	2017	[51]
**46.**	Aspirin	I, II	0–12 GPa	GO, 2D PES, PC, ZPE, TD	Quantum ESPRESSO, Phonopy	B86Bpbe, B86bPBE-XDM	2016	[52]
**47.**	Aspirin	I, II	0–5 GPa	GO, PC, IR	CRYSTAL	B3LYP-2D	2015	[53]
**48.**	Paracetamol	I, II	0–5 GPa	GO, PC, IR	CRYSTAL	B3LYP-2D	2015	[53]
**49.**	Resorcinol	α, β	0–4.5 GPa	GO, INS, PC, TD, Raman	CASTEP; CRYSTAL	WC, PBESOL, PW91, PBE, rPBE, PBE-D2, PBE TS, PBE/pob-TZVP	2015	[54]
**50.**	Glycine	α, β, γ, δ, ε	0–10 GPa	GO	Quantum ESPRESSO	PBE, revPBE, vdW-DF, vdW-DF-c09x	2012	[55]
**51.**	L-Alanine	NP	0–10 GPa	GO	Quantum ESPRESSO	PBE, revPBE, vdW-DF, vdW-DF-c09x	2012	[55]
**52.**	L-Serine	I, II, III	0–8.1 GPa	GO	SIESTA	PBE	2008	[56]
**D. Others**								
**53.**	Ammonium carbamate	α, β	0–15 GPa	GO	CASTEP	PBE TS	2022	[57]
**54.**	Chloroform (CHCl_3_)	*P*6_3_, *Pnma*	0–35 GPa	GO, Raman, PF	CASTEP	PBE TS	2020	[58]
**55.**	Croconic acid	*Pca*2_1_, *Pbcm*	0–55 GPa	USPEX, GO (compression and decompression), PC, Raman, PF, BG	CASTEP	PBE TS	2020	[59]
**56.**	Squaric acid	*P*2_1_*/m*, *I*4*m*	0–25 GPa	USPEX, GO (compression and decompression), PC, Raman, PF, BG, OP	CASTEP	PBE TS	2020	[59]
**57.**	Diisopropylammonium perchlorate (DIPAP)	P1	0–3.3 GPa	GO, Raman	DMol3	PBE, PBE Grimme	2020	[60]
**58.**	C_11_N_4_	g-C_11_N_4_, α-C_11_N_4_, d-C_11_N_4_, and β-C_11_N_4_	0 -70 GPa	GO, PC, PF, TD	VASP	PBESOL	2019	[61]
**59.**	Oxalic acid	Dihydrate, α and β	(−1.0)–12.0 GPa	GO, XRD	CASTEP	PBE	2019	[62]
**60.**	Sorbic acid	C2/c	0–8 GPa	GO, PC, Raman	CASTEP	PBE	2017	[63]
**61.**	Oxobenzene-bridged 1,2,3-bisdithiazolyl radical conductor (3a)	α, β, γ	0–13 GPa	SP, MO, BG	Gaussian; Quantum ESPRESSO	(U)B3LYP; PBE	2014	[64]
**62.**	Bis-1,2,3-thiaselenazolyl radical dimer	[1a]_2_: σ-dimer, π-dimer [1b]_2_: NP	0-13.7 GPa	GO, BG	VASP	PBE	2012	[65]
**63.**	Indole	HB and β	0–25 GPa	GO, TB	CASTEP	PBE TS	2011	[66]

Abbreviations: 2D PES—two-dimensional potential energy surface; AIM—atoms in molecules; AS—absorption spectra calculations; aiMD—ab initio molecular dynamics; BG—bandgap; CM—center-of-mass fractional position calculations; DOS—density of states; ES—excited state calculation; GO—geometry optimization; HS—Hirshfeld surface; IGM—intramolecular gradients method; INS—inelastic neutron scattering; MO—molecular orbitals; MPD—mutual penetration distance; MSST—multi-scale shock technique; NA—not applicable; NBO—natural bond orbitals; NP—not provided; OP—optical properties; PC—phonon DOS calculation; PD—phase diagram; PF—phonon frequency; QMD—quantum molecular dynamics; SCC-DFTB—self-consistent charge density functional tight binding; SP—single-point calculations; SOC—spin–orbit coupling; TB—transition barrier calculation; TD—thermodynamics; TD-DFT—time-dependent density functional theory calculations; VTST—variational transition-state theory; XRD—X-ray diffraction.

**Table 2 ijms-24-14155-t002:** Software applied in the reviewed papers in descending order of number of works.

N°	Software/Code	Basis Set	Periodic	License Type	Ref. Method	Number of Works	Ref. in This Article
1.	CASTEP	Plane-wave	3D	Academic, Commercial	[67,68]	27	[13,21,25,26,27,29,32,34,37,39,43,44,45,47,49,54,57,58,59,62,63,66,69]
2.	VASP	Plane-wave	3D	Academic, Commercial	[70,71]	11	[15,16,17,20,22,28,46,50,51,61,65]
3.	Quantum ESPRESSO	Plane-wave	3D	Free, General Public License	[72,73]	9	[14,23,35,40,41,42,52,55,64]
4.	Gaussian	Gaussian-type orbitals	Any	Commercial	[74,75]	8	[12,19,24,36,38,50,51,64]
5.	CP2K	Hybrid Gaussian-type orbitals, plane-wave	Any	Free, General Public License	[76,77]	4	[30,31,32,33]
6.	CRYSTAL	Gaussian-type orbitals	Any	Academic, Commercial	[78,79]	3	[48,53,54]
7.	DMol3	Numerically tabulated atom-centered orbitals	Any	Commercial	[80,81]	3	[36,37,60]
8.	CPMD	Plane-wave	3D	Academic	[82,83]	1	[36]
9.	DFTB+	Slater-type orbitals, numerically tabulated atom-centered orbitals	Any	Free, General Public License	[84,85]	1	[41]
10.	SIESTA	Numerically tabulated atom-centered orbitals	3D	Free, General Public License	[86,87]	1	[56]
11.	WIEN2k	FP-(L)APW + lo (the full-potential (linearized) augmented plane-wave and local orbitals)	3D	Commercial	[88,89]	1	[34]

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
