# Peer review of "Review of Applications of Density Functional Theory (DFT) Quantum Mechanical Calculations to Study the High-Pressure Polymorphs of Organic Crystalline Materials"

_ijms, 2023, doi:10.3390/ijms241814155_

Round 1

Reviewer 1 Report

The authors have made an effort to introduce the chemical aspects of density functional theory (DFT) quantum mechanical calculations to study the high pressure polymorphs of organic crystalline materials. After carefully reading the manuscript, I believe it is suitable for publication in this journal after carefully considering the comments below.

1. it is not clear to the reader why the activity of the organic materials with metal additives composites would be higher than that of the other from the DFT viewpoint. Could the authors elaborate on this?

2. There are serious grammatical errors in this manuscript. For example, the following expression in the 3rd line from the bottom on page 28 is not a complete sentence

3. In the concluding section, it is necessary to describe in more detail the prospects for the DFT.

4. An abbreviation list is required

5. Some related refs could be cited, such as Theor Chem Acc, 142 (2023) 78; Theor. Chem. Acc. 2022, 141, 68 and J. Phys. Chem. A, 2019, 123, 6751−6760.

6. The summary and future prospects can be slightly elaborated.

7. Authors need to check section dividing numbers throughout the manuscript. 

8. Authors need to supply a detailed information of feature perspectives to overcome the limitations of present approaching methods and techniques.

9. Please add conclusive summary figure at the last part of the manuscript. Perspective is also desirably illustrated.

work carefully

Reviewer 2 Report

The review by Napiórkowska et al. aims to encompass the DFT studies of organic and hybrid metal-organic compounds at high pressure. In the current state, the manuscript needs improvement and can be considered after major revision. The list of issues is given below:

1. The set of objects considered in the review needs either to be expanded significantly or the choice of the objects needs to be justified. Many inorganic and metal-organic compounds have been studied using computational methods (the list of papers is too long to provide here), which is not reflected in the considered paper. Silver fulminate is generally considered an inorganic compound. The classification of energetic materials is fine but Table 1 contains a lot of pharmaceuticals that also need to be considered in a separate section. The software in Table 2 can be grouped depending on the object for possible conclusions. E.g. some software might perform better for a certain class of compounds but never used for other classes.

2. The crystal structure prediction is not limited to USPEX method. The list of the available codes for CSP can be found in the reviews, e.g. in [10.1039/D1CE01564H; 10.1039/C8FD00033F]. The Crystal Structure Prediction blind tests (six completed [10.1107/S2052520616007447], the results of the seventh are yet to be published: https://www.ccdc.cam.ac.uk/discover/news/conclusions-of-the-7th-crystal-structure-prediction-blind-test/). Although USPEX is highly popular (mainly since it is free), the results of this method are ranked lower compared to other methods. At least the highly cited results obtained using GRACE [10.1038/ncomms8793; 10.1021/jacs.9b06634] need to be presented in the review.

3. Table 1 contains many typos and questionnable choices. CRYSTAL and CRYSTAL09 are the same package, and Gaussian is not a software for periodic computations (the PBC module does not fully use the crystal symmetry). CrystalExplorer uses HF/DFT data of isolated molecules from Gaussian to work properly, so its categorization as DFT software is questionnable. Inclusion of the works that study the systems only at 0 GPa also needs justification.

Reviewer 3 Report

The paper under consideration collects a lot articles on the DFT calculations on the DFT studies on the high-pressure polymorphs of organic crystal. This broad and fascinating topic is of high interest for the broader community of chemists, physists and pharmacists and from the point of view of the community of quantum chemists represents an interesting challenge. Thus, an extensive review on the subject is by all means welcome. My  problem with the article however is that with all relevant papers quoted the text is fairly unfocused and the information on particular problems/effects is hard to find. Already the titles of the chapters shows some problems with organising of the paper. It beginns with the introduction and brief theoretical background and the large chapter "Organic materials with metal additives". In fact the Table 1 shows a number of papers on pure organic molecules with few coordination compounds and few metallorganic compounds. The authors used the term  "organometallic materials" (line 114) while the discussed molecules show hardly the metallorganic (Metal-carbon or metal-silicon bonds are not present) character. OK, the next chapter is called "High energy materials" with following "Kinetic barriers", "Crystal  structure USPEX algorithm" (where a particular package is discussed), "Vibrational spectra- signal assignment", " Phonon frequency calculations" (we have two chapters dealing essentialy the same problem of calculating the vibrations), then "Determination of pressure induced phase transition conditions" with an rather cryptically titled chapter "Properties" (like if it were no properties that have been discussed before) and equally enigmatically titled "No pressure-induced phase transition". Finally "10. Anisotropic compression" is discussed, followed by "5. Conclusions". Why 5?
So we have introduction, theoretical background, a chapter on different molecules, then kinetic is discussed, particular code is mentioned, the vibrational properties are discussed to finally devote a chapter to a particular problem of anisotropic compression. It does look like a well structured text, does it? In my opinion one purpose of a review article is to give an introduction to the field to the newcomers. It has to be organised in a way that suits this very purpose. It is not enough to write up everything the authors know about the published papers.
My suggestion to authors is to reorganise the paper in the following way:
-After the introduction and the theoretical background given the different molecular crystals that have been modelled shall be listed, much alike it is already done with high-energy materials as a special class. Wouldn't a separate subchapter on pharmaceutically relevant compounds a good idea, taking into account the importance of the polymorphs for the drug design?
-The next chapter shall include the results dealing with fundamental properties like structure, enthalpy and vibrations. With vibrations being the essential part of the crystal entropy the derivation of the thermodynamic properties. It would be essential at this point (we still assume the readers might an introduction to the field) that the solid-state normal mode calculations are the another level of jobs compared to the geometry/crystal cell parameters ones. Finally, the calculations of phonon properties would come as the consequence of the above mentioned problems.

The next chapter may be devoted to the particular problem like these currently given in chapters 9-10.

What I would also suggest to authors is to give some insight into problems how different levels of theory imply the reliability of the results with the cost being computational time. In other words a typical dillema of quantum chemistry methods that are "either to true to be good or to good to be true". Finally, it is surely possible how the choice of exchange-correlation functional and basis set influences the results. Few example of this effect for one modelled system may be given in the theoretical part.

I advise the authors to reconsider the titles of the chapters and subchapters. I found also few point where the English does not sound too English. Particularly "what is more" (line 280) seems to be not particularly well sounding in this context.

Round 2

Reviewer 1 Report

accept

Reviewer 3 Report

I think the paper in its current form looks significantly better. I recommend the publication of it.